



# On optimization of calibrations of a distributed hydrological model with spatially distributed information on snow

Dipti Tiwari[1], Mélanie Trudel[1], and Robert Leconte[1]

[1]Université de Sherbrooke, Département de génie civil et de génie du bâtiment, 2500 Bd de l'Université, Sherbrooke, QC J1K 2R1,Canada

**Correspondence:** Dipti Tiwari (dipti.tiwari@usherbrooke.ca)

**Abstract.** In northern cold-temperate countries, a large portion of annual streamflow is produced by spring snowmelt, which often triggers floods. It is important to have spatial information about snow parameters such as snow water equivalent (SWE), which can be incorporated into hydrological models, making them more efficient tools for improved decision-making. The future Terrestrial Snow Mass Mission (TSMM) aims to provide high-resolution spatially distributed SWE information; thus,

spatial SWE calibration should be considered along with conventional streamflow calibration for model optimization since the overall water balance is often a key objective in the hydrological modelling. The present research implements a unique spatial pattern metric in a multi-objective framework for calibration approach of hydrological models and attempts to determine whether raw SNODAS data can be utilized for hydrological model calibration. The SPAtial Efficiency (SPAEF) metric is explored for spatially calibrating SWE. The HYDROTEL hydrological model is applied to the Au Saumon River Watershed

($\backsim$1120 $km^2$) in Eastern Canada using MSWEP precipitation data and ERA-5 land reanalysis temperature data as input to generate high-resolution SWE and streamflow. Different calibration experiments are performed combining Nash-Sutcliffe efficiency (NSE) for streamflow and root-mean-square error (RMSE), and SPAEF for SWE, using the Dynamically Dimensioned Search (DDS) and Pareto Archived multi-objective optimization (PADDS) algorithms. Results of the study demonstrate that multi-objective calibration outperforms sequential calibration in terms of model performance. Traditional model calibration in-

volving only streamflow produced slightly higher NSE values; however, the spatial distribution of SWE could not be adequately maintained. This study indicates that utilizing SPAEF for spatial calibration of snow parameters improved streamflow prediction compared to the conventional practice of using RMSE for calibration. SPAEF is further implied to be a more effective metric than RMSE for both sequential and multi-objective calibration. During validation, the calibration experiment incorporating multi-objective SPAEF exhibits enhanced performance in terms of NSE and Kling-Gupta Efficiency (KGE) compared

to calibration experiment solely based on NSE. This observation supports the notion that incorporating SPAEF computed on raw SNODAS data within the calibration framework results in a more robust hydrological model.

## 1 Introduction

Cold temperate countries like Canada are characterized by substantial spring runoff, where streamflow is generated or amplified by snowmelt. Indeed, spring snowmelt is one of the predominant hydrological event influencing the annual water budget of





high latitude watersheds (DeWalle and Rango, 2008; Buttle et al., 2016). Rapid melting of snow can result from a combination of high temperatures and rainfall over frozen ground resulting in heavy runoff, which will exacerbate flooding. A reliable flow forecast system, therefore, should provide inflow forecasts using hydrological models that can simulate the complex processes occurring in the watershed. These models can mimic the relationship between inflow and outflow in a watershed provided that they have adequate meteorological data (precipitation, temperature, relative humidity, wind speed and direction) and geograph-

ical data (digital elevation model, land cover and soil maps) (Singh and Woolhiser, 2002). Distributed hydrological models are capable of representing the spatial variability of hydrological processes and state variables within a watershed, which cannot be obtained by lumped models (Markhali et al., 2022).

Snow distribution, together with the frequency, duration and intensity of spring snowmelt, influences extreme runoff events (Marsh et al., 2008), soil water storage capacity, subsurface flow (Roach et al., 2011; Frampton et al., 2013; Jafarov et al.,

2018) and groundwater recharge (Mohammed et al., 2019; Ala-Aho et al., 2021). Owing to the dominant control of snow on the water balance during spring, the spatial distribution of snow has received much attention (Hiemstra et al., 2002; Woo and Young, 2004). Spatial variability of snow cover results from various processes that occur across different spatial scales (Clark et al., 2011). At the watershed scale, it is affected by meteorological variables (e.g., temperature, precipitation, evaporation) and by elevation, topography and vegetation, while at the hillslope-scale it is governed by processes such as drifting, trapping

of snow, and the nonuniform snow unloading by the forest canopy (Hojatimalekshah et al., 2021). Methods that are available for analyzing spatial variation of the snowpack have their limitations and incorporating them into hydrological models is difficult. For example, interpolation techniques (Harshburger et al., 2010) for generating snow water equivalent (SWE) maps at the watershed scale require a surface network of ground SWE measurements, which are frequently limited in number or are absent. Passive micro-wave sensors provide appropriate information on snow in areas with simple topography, but they do not

work properly in hilly terrain, typically underestimating SWE (Wrzesien et al., 2017). Moreover, their low spatial resolution requires appropriate downscaling approaches to bring the information to the watershed scale. GlobSnow (Luojus et al., 2020), which is currently available as a global snow monitoring system, integrates passive microwave remote sensing data from multiple satellites to provide near-real-time information on snow cover extent at a global scale. This valuable resource offers insights into the spatial distribution of snow cover across diverse regions. On one hand, and despite its easy accessibility,

GlobSnow exhibits relatively coarse resolution, thereby limiting its ability to capture fine-scale details. On the other hand, SNODAS (SNOw Data Assimilation System) (NOHRSC, 2004) represents a subsequent advancement that integrates passive microwave remote sensing data, ground measurements and weather data to generate gridded snow data sets at a finer resolution. The incorporation of various data sources and assimilation techniques into SNODAS enhances the quality and reliability of snow data, enabling more accurate hydrological modelling and facilitating rigorous scientific investigation. The future

Terrestrial Snow Mass Mission (TSMM) is a low-cost, low-mass, spaceborne Ku-band SAR system that is being developed by the Canadian Space Agency (Derksen et al., 2021), and which is expected to provide high spatial resolution SWE data. In preparation for this mission, together with existing remotely sensed SWE products such as SNODAS, it is essential that the possibilities of these products be investigated for operational environmental prediction, such as spring flood forecasting.



SNODAS is a modelling and data assimilation system that had been developed by NOHRSC (National Operational Hydrologic Remote Sensing Center). It uses a physically-based mass and energy balance snow model that assimilates automated ground, airborne measurements and satellite data (Zahmatkesh et al., 2019) to provide accurate estimates of snow cover and associated parameters for hydrological modelling and analysis. The dataset includes snowpack properties such as depth and snow water equivalent (SWE), covering parameters such as liquid precipitation, snowmelt runoff, temperature, and more (Barrett, 2003). It has provided daily, gridded data at a 1 km x 1 km spatial resolution for the USA since 2004 and for southern Canada since 2009.

Researchers used bias-corrected SNODAS SWE data in hydrological modelling, given that it provides more accurate and reliable input data for the models whereas raw SNODAS data may contain errors and biases. These errors and biases can adversely affect the accuracy of hydrological models that rely upon SNODAS data as input. A study by Clow et al. (2012) concluded that SNODAS provides reasonably true estimates of SWE in forested areas and can be used as observed data to calibrate hydrological models for moderate- to large-watersheds and for estimating runoff forecasts. Zahmatkesh et al. (2019) demonstrated that bias-correcting SNODAS SWE products, which align with the cumulative distribution function of interpolated SWE, improved the accuracy of the SNODAS products and enhanced model performance in simulating peak values during hydrological modelling and streamflow simulation. Despite the high uncertainty that is associated with SNODAS estimates in eastern Canadian watersheds, they remain valuable in regions with limited snow stations. Bias correction of SNODAS data may not be possible in situations where there is a lack of ground-based measurements or observations with which to compare the SNODAS data. Given the unavailability of observed SWE data in the study area, Leach et al. (2018) applied a bias correction technique, which was originally used for SNODAS snow depth data, to adjust the SNODAS SWE dataset, assuming a comparable level of relative bias between snow depth and SWE based upon their close relationship with the observations. This paucity of data compels researchers to explore new methods that allow the use of SNODAS data without relying upon traditional bias correction techniques.

This study addresses the need to investigate the feasibility and efficiency of employing raw SNODAS data for calibrating hydrological models without bias correction, utilizing diverse objective functions, and evaluating the adequacy of the uncorrected SNODAS data in generating satisfactory results. The current research aims to bridge this gap by examining potential approaches that would enhance the reliability and accuracy of model outputs when working with raw SNODAS data, thereby contributing to the advancement of hydrological modelling techniques.

Distributed hydrological models, despite their capacity to mimic the spatial distribution of hydrological state variables and fluxes, continue to be used mainly for their temporal characteristics of the aggregated streamflow variable (Demirel et al., 2013; Schumann et al., 2013). Snow significantly influences the seasonal characteristics of streamflow, thereby affecting other hydrological processes, such as erosion, water supply, and flood forecast. Given the importance of accurately representing snow processes in snow-dominated watersheds, hydrological model calibration should not solely focus on streamflow but should also account for snow dynamics (Troin and Caya, 2014; Hanzer et al., 2016; Tuo et al., 2018). Snow-related observations are frequently utilized to improve hydrological model performance through calibration experiments (Roy et al., 2010; Parajka and Blöschl, 2008; Di Marco et al., 2021). Hydrological model calibration in snow dominated regions is complex because calibra-





tion based on snow parameters does not necessarily lead to optimal parameters for streamflow and vice versa . Commonly used

hydrological models can accurately replicate streamflow observations by calibrating them with respect to streamflow alone. However, they might struggle to properly capture state variables, particularly related to the snowpack (Duethmann et al., 2014; Casson et al., 2018; Liu et al., 2020; Thornton et al., 2021). For example, the models are usually calibrated using streamflow as the sole hydrological variable that is being simulated by the models, leaving other state variables, such as snow and soil mois-ture unused in the calibration procedure. This is because spatially distributed observations traditionally have been difficult to

obtain. To improve the reliability of hydrological models, it is critical to assess the simulated patterns of model state variables against spatially distributed observations. Ensuring that the models can accurately reproduce the spatio-temporal dynamics of all relevant hydrological processes, leading to accurate streamflow forecasts (Kirchner, 2006). The increased availability of remotely sensed observations has opened new doors in hydrological model calibration (e.g., snow cover, Terink et al. (2015); land surface temperature, Stisen et al. (2021)). The objective of this research is to investigate strategies that include distributed

snow information in the calibration of a distributed hydrological model, with a specific focus upon understanding the influence of different objective functions on the calibration process. The selection of an appropriate set of objective functions is crucial, as it introduces trade-offs among model parameters that influence overall calibration performance and model accuracy.

A number of temporal metrics are available for comparing simulated versus observed hydrographs, but spatial distribution comparison is occasionally used (Rees, 2008; Koch et al., 2018). Moreover, the spatial information that is contained in the

observed data is not optimally utilized, given that the model parameters are constrained against spatial observations. Spatial metrics generally assess grid-to-grid correlation, and deviations such as bias and Pearson's correlation coefficient R. Demirel et al. (2018) developed a new SPAtial EFficiency metric (SPAEF) with histogram matching to compare raster maps. The metric was later modified to include three different components, i.e., histogram intersection, variance and correlation, which offer bias-insensitive pattern information. This new objective function SPAEF is used in this study to calibrate SWE spatially.

To evaluate the added-value of calibration taking into account spatial variation of the snow, different calibration experiments are performed using search algorithms, DDS (Dynamically Dimensioned Search) (Tolson and Shoemaker, 2007) for single objective function, and PADDS (Pareto Archived Dynamically Dimensioned Search) (Asadzadeh and Tolson, 2013) for multi-objective functions, with the distributed hydrological model HYDROTEL (Fortin et al., 2001) to optimize model performance. DDS is a global optimization algorithm for automatic calibration of hydrological models (Tolson and Shoemaker, 2007),

which optimizes one objective function at a time without requiring algorithm parameter tuning. In contrast, PADDS is a multi-objective dynamic algorithm that uses DDS as a search engine to optimize multiple objective functions by perturbing one non-dominated solution every iteration and archiving all non-dominated solutions throughout the search. PADDS algorithm offers multiple selection metrics function for example Random selection, Crowding distance (CD), Hypervolume Contribution (HVC) and Convex Hull Contribution (CHC). To learn more about these metrics please refer to Jahanpour et al. (2018) and Tolson

and Jahanpour (2018). In this study, the HVC selection metric was chosen for the PADDS multiobjective calibration method. The effectiveness of this metric has been demonstrated in previous research (Asadzadeh and Tolson, 2013). When used with multi-objective functions the HVC measures the increase in hypervolume achieved by adding a solution to an existing set and explores the Pareto front, which represents a range of optimal trade-off solutions, by dynamically adjusting the dimensionality





of the search space. Solutions with higher HVC values are prioritized to improve the coverage of the Pareto front, enhancing
the overall quality of the calibration process. In this study, DDS is used when only one objective function is used at a time
for calibration, i.e., first three experiments, while PADDS is used to optimize the model with two or more objective functions
simultaneously. This article focuses on utilizing the spatially distributed snow information to calibrate the model to achieve
better model performance with respect to both SWE and streamflow. The objective functions that are used during calibration are
NSE (Nash-Sutcliffe Efficiency) for streamflow, RMSE (Root-Mean-Square Error) for optimization of SWE over each RHHUs
(relatively homogeneous hydrological units), and SPAEF (spatial efficiency metric) for spatially distributed optimization of
SWE. As NSE cannot be spatialize but can only be calculated on the average SWE over the watershed, here we preferred
RMSE as an objective function to calibrate SWE, because it can be applied to each RHHU (spatially) and not only on the
average SWE. In addition, Kling-Gupta Efficiency (KGE) is utilized for validation purposes. The supplementary material of
this article presents a thorough analysis of the trade-offs between different objectives: $NSE_Q$ and $RMSE_{SWE}$, and $NSE_Q$ and
$SPAEF_{SWE}$. Readers are encouraged to refer to the supplementary section for an in-depth exploration of these trade-offs and
gain valuable insights about the performance of the algorithm with respect to the mentioned objectives.

## 2 Experimental Setup

### 2.1 Study site:

The Au Saumon watershed covers an area about 1120 $km^2$ and is located in the Éstrie region of Québec (eastern Canada). It is
a snow-dominated watershed that is subject to different climate conditions (summers tend to be warm and humid, with heavy
rainfall; winters are cold and snowy; spring is the snowmelt season, with frequent rainfall; and autumn is drier with cooler
temperatures). The watershed has mostly a rolling topography. Elevation ranges from 240 m at the watershed outlet, further
rising upto 1100 m in the southern and eastern portions due to presence of Mont Mégantic (Figure-1). This variation in elevation
affects the spatial distribution of snow during winter, given that it affects temperatures. While rising to higher elevations, the
temperature drops gradually, causing different patterns of melting and freezing in flat and hilly places. Vegetation cover in
the watershed consists mainly of forests, i.e., 7% conifer, 32% deciduous, and 45% mixed forest. The remaining land cover
consists of 9% crop lands, 2% urban, and about 5% wetlands, shrub lands and grasslands (Figure-1). Winter climate dominates
this watershed with snow cover usually forming in late November or early December and ending in April. Average minimum
and maximum air temperatures during winter are -12 $°C$ and -5 $°C$, respectively. Annual precipitation ranges from 1000 mm
to 1200 mm (https://climate.weather.gc.ca/index_e.html), with about one-third falling as snow (Bergeron et al., 2014). The
parent material yields sandstone, limestone, and shale types of soils (silty-loams) (https://sis.agr.gc.ca/cansis/) (Seiller et al.,
2012). There is one streamflow station (030282) as shown in Figure (1) (https://www.cehq.gouv.qc.ca/hydrometrie/historique_
donnees/), with a drainage area of 769 km2. There are no SWE data observation stations within the watershed; however,
stations at Milan (station number:302060) and Bury (station number:302100) are located in close proximity to the watershed.
These stations provide point data for SWE measurements (info-climat MELCCFP, 2020).

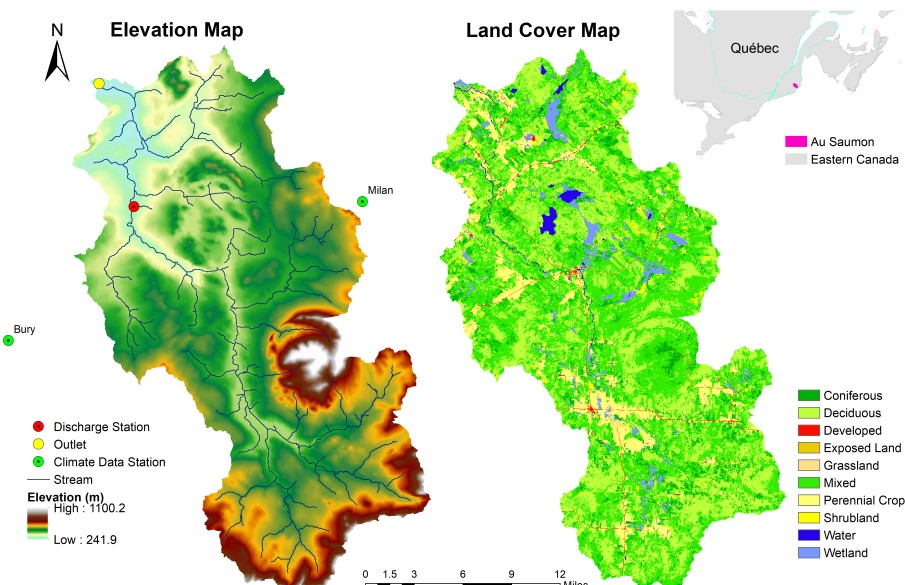

**Figure 1.** Au Saumon Watershed location, elevation, stream network, streamflow outlet and land cover

## 2.2    HYDROTEL Model:

The model that is selected for this study is a process-based, continuous, distributed hydrological model HYDROTEL, which was developed by Fortin et al. (1995). The model has been used in a number of watersheds to study different hydrological processes, including snow water equivalent (Turcotte et al., 2007; Oreiller et al., 2014; Fossey et al., 2016; Augas et al., 2020)
and flow forecasting (Turcotte et al., 2004; Abaza et al., 2014, 2015). HYDROTEL serves as the base model for Quebec's operational flow forecasting system. This model discretizes the watershed in several simulation units that are referred to as relatively homogeneous hydrological units (RHHUs) and river reaches (Turcotte et al., 2001). The characteristics of each RHHU depend upon land cover, soil types and topography (Rousseau et al., 2011). HYDROTEL can be simulated on a daily basis or in 3-hour time steps. In our study, the Au Saumon Watershed is subdivided into 205 RHHUs based upon the spatial
distribution of land use, land cover, soil properties, slope and elevation. The mean surface area of the RHHUs is 4.5 $km^2$.

HYDROTEL is composed of different modules, which run consecutively. The snow module uses a single-layer structure and is based upon a mixed degree-day / energy balance hybrid approach. Snowpack characteristics (water equivalent, thickness, mean density, liquid water content, thermal deficit, temperature) are simulated using a modified energy budget approach that was developed by Riley et al. (1972). Empirical relationships are used to produce air-snow and ground-snow interface melt,
albedo evolution, compaction, and the liquid water that is retained by the snow cover (Turcotte et al., 2007). The other modules selected in this study are the Thiessen polygon method for interpolation of meteorological variables, the Rankine method for soil temperature, the Thornthwaite equation for potential evapotranspiration and a three-layer model (BV3C) for the vertical





Table 1. .Lower and Upper Bounds of the parameters and Initial parameter values of base model

| Parameters and their upper and lower bound (DDS/PADDS Algorithm) | | | | Base Model |
|---|---|---|---|---|
| Parameter Name | Lower Bound | Initial value | Upper Bound | Parameter Value |
| 1. Base refreezing temperature (mm/d) | -3 | -1.65 | 2 | -0.88 |
| 2. Temperature threshold for melt- Coniferous (°C) | -4 | 0.1 | 4 | 0.61 |
| 3. Temperature threshold for melt- Deciduous (°C) | -4 | 0.48 | 4 | -0.08 |
| 4. Temperature threshold for melt- Open (°C) | -4 | 0.78 | 4 | 2.00 |
| 5. Melt factor for coniferous forests (mm/d per °C) | 2 | 9.62 | 15 | 7.83 |
| 6. Melt factor for deciduous forests (mm/d per °C) | 2 | 8.27 | 15 | 9.69 |
| 7. Melt factor for open areas (mm/d per °C) | 2 | 11.42 | 15 | 5.00 |
| 8. Multiplication factor for PET | 0.7 | 0.93 | 1.5 | 1.16 |
| 9. Depth of the first soil layer (m.) | 0.01 | 0.11 | 0.2 | 0.02 |
| 10. Depth of the second soil layer (m.) | 0.1 | 0.45 | 1.5 | 0.42 |
| 11. Depth of the third soil layer (m.) | 1 | 4.97 | 7 | 1.00 |

water budget in the soil column. The output flow is modeled using the kinematic wave equation, and the model is simulated on a daily basis.

Of all parameters that are available in different modules, a subset of 11 parameters was selected for model calibration, as listed in Table-1. They include seven snow-related parameters, three parameters for soil layer thickness of the 3-layer soil column, and one parameter for converting potential evapotranspiration (PET) into actual evapotranspiration. The snow-related parameters affect the evolution of SWE in each RHHU, while the other parameters affect runoff that is generated by the model. These parameters are selected based on sensitivity analyses done in previous studies on different watersheds (Bouda et al., 185    2014; Huot et al., 2019; Lucas-Picher et al., 2020).

### 2.3    Meteorological and streamflow data

For this study, HYDROTEL is forced with spatially distributed meteorological precipitation and minimum and maximum temperature data. For precipitation, the data that are used are from MSWEP (Multi-Source Weighted-Ensemble Precipitation), which is a reanalysis product combining satellite data, gauge data and numerical weather model output. MSWEP is available 190    globally on daily and 3-hour bases from 1979 until today (http://www.gloh2o.org/mswep/). Grid cell resolution of the data is 0.1 degrees, which is about 10 km at the equator. Overall, MSWEP offers superior performance compared to other data sets (e.g., ERA-5 interim, ERA-5, CHIRP, and others) (Beck et al., 2017; Xiang et al., 2021). ERA5-Land dataset is used for maximum and minimum air temperature. ERA5-Land has been produced by numerical integrations of the global high-resolution ECMWF land surface model with ERA5 climate reanalysis with elevation correction (Muñoz-Sabater et al., 2021) 195    with grid cell resolution of 0.1 degrees (about 9 km native). The gridded data for both MSWEP precipitation data and ERA5-Land temperature data that were used in our study range from October 2000 to September 2020 and cover 45°10' N to 45°50'



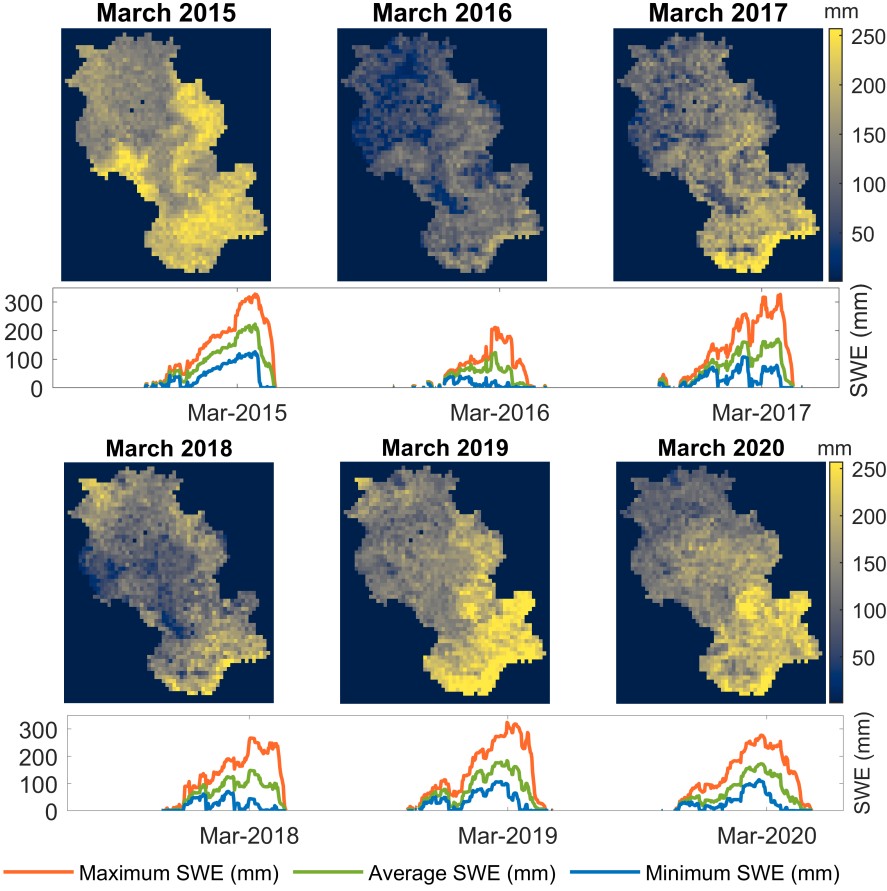

**Figure 2.** Spatially distributed average SWE for March, together with maximum, minimum and mean SWE for years 2015 to 2020

N and 71° W to 71°30' W. Observed daily streamflow data that were used for model calibration originate from the Ministère de l'Environnement, de la Lutte contre les changements climatiques, de la Faune et des Parcs station 030282 (NAD83; 45° 34' 48" // -71° 23' 6"). The streamflow station is Au Saumon (030282), which is located 1.9 km upstream from the watershed
outlet (see Figure-1). The availability of streamflow data is from 1974 and onward.

SWE data from SNODAS are used as observed data for model calibration. Figure 2 represents the spatial distribution of average SWE for the month of March for the period 2014-2020 as obtained from SNODAS, together with the temporal maximum, minimum and average of SWE across the Au Saumon Watershed. It is clearly indicated in the figure that the spatial distribution varies from year to year, even when the spatially averaged values of SWE are in close agreement with each other.
This is the case for years 2017 and 2018 and for years 2019 and 2020, where a difference of about 10 mm in average SWE in March is observed. In this study, March is selected for the spatial calibration experiment, given that the most accumulated snow during the winter season typically occurred in this month. For our study, model calibration was performed for the period





spanning 2014 to 2020, given that SNODAS data for the Au Saumon region are available from 2014 onwards, while model validation of streamflow only was conducted for the period ranging from 2001 to 2013.

## 3 Methodology

Hydrological models are typically calibrated using a single objective function, which focuses on only one aspect of the hydrological features, i.e., streamflow (Tolson and Shoemaker, 2007). As hydrological models have multiple outputs, this draws attention towards using a multi-objective approach to explore the various hydrological feature information that is stored in hydrological data and, thus, moving toward a multi- objective model calibration. Optimizing two or more objective functions simultaneously provides a better overall calibrated hydrological model (Adeyeri et al., 2020). The objective functions selected for this study and the optimization algorithms are now presented in the next section.

### 3.1 Objective Functions used in the study

In this study, the NSE, RMSE and SPAEF are employed as objective functions for calibration, while the KGE is used for validation of the model. By using these three objective functions, various calibration experiments were designed to assess the strengths and weaknesses of each experiment in evaluating overall model performance.

NSE (Equation-1) is used to evaluate the predictive skill of hydrological models. It basically compares error variance of the simulated time-series, which are typically flow data, with the magnitude of the observed time-series.

$$NSE = 1 - \frac{\sum_{t=1}^{T} \left(Q_o^t - Q_m^t\right)^2}{\sum_{t=1}^{T} \left(Q_o^t - \bar{Q}_o\right)^2} \tag{1}$$

where $Q_o$ is observed discharge, $\bar{Q}_o$ is the mean of observed discharges, and $Q_m$ is modeled discharge.

The efficiency ranges from $-\infty$ to 1. The value of NSE is maximized during model calibration. For a perfect hydrological model, the estimated error variance is zero and the resulting NSE is 1, whereas NSE is 0 when an estimated error variance is equal to the variance of the observed time series. If the NSE value is close to 1, it suggests that the model will have more predictive skill. Krause et al. (2005) specified that drawback of using NSE as objective function is that it uses squares values to calculate difference between the simulated values and observed values which results in overestimation of larger values while underestimation of smaller values. So, this results in underestimation of flow during low flow conditions, yet it is the best objective function to replicate the peak runoff discharge matching on a hydrograph. In this study, the NSE is calculated between the observed flow at discharge station and the simulated flow at the corresponding RHHU.

RMSE (Equation- 2) is the standard deviation of prediction errors. In other words, it measures how far do the data points move away from the regression line. The smaller the value of RMSE, the better is model performance. RMSE has been used widely for SWE optimization (Larue et al., 2018; Follum et al., 2019; Koch et al., 2019; Sleziak et al., 2020; van Tiel et al., 2020; Yang et al., 2020).


$$RMSE = \sqrt{\sum_{t}^{N} \sum_{i}^{RHHU} \frac{(SWE\_HYDROTEL_{\text{RHHU},t} - SWE\_SNODAS_{\text{RHHU},t})^2}{N \cdot \text{RHHU}}} \qquad (2)$$

Where $t$ is the time step, and $N$ is the number of time steps when HYDROTEL or SNODAS have a non-zero SWE value. $SWE\_HYDROTEL$ is the SWE computed by HYDROTEL for each RHHU and time step, and $SWE\_SNODAS$ is the
average SWE of SNODAS over the RHHU for each time step. In this study, the RMSE is calculated for each of the 205 RHHUs within the Au Saumon watershed. The spatial RMSE values are calculated by comparing the SNODAS SWE and the simulated Hydrotel SWE. Subsequently, the RMSE is minimized to enhance the model's performance.

SPAEF (Equation-3) is a metric that is used to assess the spatial performance of a model, as opposed to NSE and RMSE, which are used to evaluate temporal model performance. SPAEF has been developed to calibrate distributed hydrological
models so as to better represent the spatial variability of hydrological processes (Demirel et al., 2018; Koch et al., 2018; Demirel, 2020). In our study, SPAEF is used for assessing spatial patterns of SWE. SPAEF is calculated according to the following equation:

$$SPAEF = 1 - \sqrt{(A-1)^2 + (B-1)^2 + (C-1)^2} \qquad (3)$$

where

$A = \rho(\text{obs}, \text{sim})$      $B = \left(\frac{\sigma_{\text{sim}}}{\mu_{\text{sim}}}\right) / \left(\frac{\sigma_{\text{obs}}}{\mu_{\text{obs}}}\right)$      $C = \frac{\sum_{j=1}^{n} \min(K_j, L_j)}{\sum_{j=1}^{n} K_j}$

$A$ is the Pearson correlation coefficient between the observed and simulated pattern, $B$ is the fraction of the coefficient of variation representing spatial variability, $C$ is the histogram intersection for the histogram $L$ of the simulated pattern and the given histogram $K$ of the observed pattern, each containing n bins. The value of SPAEF ranges between $-\infty$ and 1. A SPAEF value equal to 1 means that the simulated pattern perfectly matches the observed pattern, while a value of 0 means that there is
no agreement between the predicted pattern and the observed pattern, which indicates that the model's predictions are entirely inaccurate and do not align with the observed data. An advantage of SPAEF is that it equally balances three distinct individual metrics (A, B and C above) that individually would not appropriately characterize spatial patterns. For example, Koch et al. (2018) show good correlations may occur between observed and simulated patterns, while a visual interpretation of the patterns suggests this is not the case. Using a multiple-component metric, such as SPAEF, helps disentangling such inconsistencies.
Koch et al. (2018) used SPAEF to calibrate the meso-scale Hydrological Model (mHM) for spatial distributions of actual evapotranspiration (AET). The study highlighted the importance of incorporating spatial observations in model calibration, since different ET patterns were obtained for similar simulated streamflow time-series, depending upon the objective function that was used in the calibration process.

SPAEF formulation is inspired by Kling–Gupta efficiency (Equation-4) that is characterized by equally weighted components
of variability, correlation and bias (Gupta et al., 2009), and is used frequently to evaluate streamflow simulations. In this study,





SPAEF is used for calibrating the HYDROTEL model with respect to the spatial distribution of SWE. Given that Au Saumon is a snow-dominated watershed during winters and the maximum snow that is accumulated during the month of March, it was selected for spatial calibration. First, the gridded SWE value from SNODAS is taken and the average of SWE per grid is defined for March. In this case, it produced 60*58 (spatial) distributed SWE values for each calibration year. The SWE that is

simulated by HYDROTEL for the same month is transformed into same 60*58 (spatial) distributed SWE values. These spatial patterns can then be calibrated using SPAEF.

$$KGE = 1 - \sqrt{(R-1)^2 + \left(\frac{\sigma_{sim}}{\sigma_{obs}} - 1\right)^2 + \left(\frac{\mu_{sim}}{\mu_{obs}} - 1\right)^2} \qquad (4)$$

KGE is calculated as Equation-4, where $R$ is the Pearson correlation coefficient between observed and simulated streamflow time-series, $\sigma_{obs}$ is the standard deviation in observations, $\sigma_{sim}$ the standard deviation in simulations, $\mu_{sim}$ is the simulated

mean streamflow, and $\mu_{obs}$ is the observed mean streamflow. KGE has been used to assess overall model performance for the various calibration scenarios that were investigated in this study.

### 3.2 Proposed Calibration approach

Seven different experiments were set up, with each experiment being characterized by a unique combination of objective functions and calibration strategy (see Table 2). All calibration experiments were performed on the base model, which is

defined as model that was ran with some random value without prior knowledge based upon previous studies done on similar catchments. Initial parameter values of HYDROTEL (the base model) are presented in Table 1. HYDROTEL was calibrated over a 6-year period, from October 2014 to September 2020. Prior to model calibration, a 2-year warm-up period was set up to avoid any effects of initial model conditions on its results during calibration. Validation extends from 2001 to 2014. In addition to data availability, one important aspect for choosing the calibration period was to include winter seasons that

were characterized by low, average and high SWE values. Winters in 2019 and 2020 especially were considered as high- and low-winter seasons, with corresponding basin-averaged SNODAS SWE values of 156.05 mm and 86.26 mm, respectively, at the onset of the spring melt season. The number of iterations also was fixed (1000) for each calibration for comparison. Lower and upper bounds of the parameters that were used in all calibration experiments are presented in Table 1.

Table 2 presents all calibration experiments that were performed in this research. The first calibration experiment, hereafter

denoted as the Standard Experiment, refers to the traditional calibration process that was based upon maximizing NSE$_Q$, which was calculated with simulated and observed streamflow time-series. In this experiment, all 11 parameters that are listed in Table 1 are optimized using the DDS algorithm. Experiments 2 and 3 consist of adding SWE information during the calibration procedure. This is done by sequentially calibrating HYDROTEL, first by adjusting the snow-related parameters (parameters 1 to 7, see Table 1) to minimize RMSE$_{SWE}$ (Experiment 2) or to maximize SPAEF$_{SWE}$ (Experiment 3) using DDS, after which

the runoff-related parameters (8 to 11) are adjusted with NSE$_Q$ for streamflow, again using DDS, while the snow parameters are left unchanged. Experiment 4 and 5 consists of adding both streamflow and SWE information once with average information of SWE (Experiment 4) and once with spatial information of SWE (Experiment 5) in the calibration procedure. This was achieved by maximizing NSE$_Q$ and minimizing RMSE$_{SWE}$ at once (Experiment 4), and by maximizing both NSE$_Q$ and SPAEF$_{SWE}$





**Table 2.** Calibration experiments with their corresponding objective functions that were used

| Objective Function used⟶ | NSE$_Q$ | RMSE$_{SWE}$ | SPAEF$_{SWE}$ |
|---|---|---|---|
| Calibrated Parameter ⟶ | Streamflow | Average SWE | Spatial SWE |
| Experiment- 1 (Standard) | ✓ | | |
| Experiment- 2 (Sequential) | ✓* | ✓* | |
| Experiment- 3 (Sequential) | ✓* | | ✓* |
| Experiment- 4 (Pareto) | ✓ | ✓ | |
| Experiment- 5 (Pareto) | ✓ | | ✓ |
| Experiment-6 (Pareto sequential) | ✓* | ✓* | ✓* |
| Experiment- 7 (Pareto Front) | ✓ | ✓ | ✓ |
| ✓* Streamflow is calibrated with NSE sequentially after optimizing SWE with either RMSE or SPAEF. | | | |

(Experiment 5) using PADDS. Experiment 6 consists of adding both spatial and average SWE information in the calibration
procedure. This is done by sequentially calibrating HYDROTEL, first by adjusting the snow-related parameters to minimize
RMSE$_{SWE}$ and maximize SPAEF$_{SWE}$ together using PADDS, after which NSE$_Q$ for streamflow was maximized while the
snow parameters are left unaltered. In Experiment 7, all 11 parameters are optimized while maximizing NSE$_Q$ for streamflow
and SPAEF$_{SWE}$ for spatial information of SWE and minimizing RMSE$_{SWE}$ for average information of snow.

## 4 Results

Values of the objective functions for each of the calibration/validation experiments are summarized in Table 3. The values of
objective functions NSE$_Q$, RMSE$_{SWE}$ and KGE$_Q$ corresponding to the base model are 0.630, 48.83 mm and 0.806, respec-
tively. Using the same base model would be helpful in evaluating whether the calibration performed is adequate for the model's
performance. When the model is calibrated with respect to SWE (for both average and spatial calibration), parameters from 1
to 7 are calibrated; when calibrating with respect to streamflow, all the 11 parameters are considered.

The first experiment is standard practice for calibrating hydrological models. The simulated streamflow in the Standard
Experiment generally follows the same temporal pattern for all calibrated years as does the observed streamflow, but the model
has some difficulties in capturing peak streamflow (Figure 3). More specifically, the model generated more streamflow during
winter 2020 (Jan to April 2020). The observed discrepancy between the simulated and observed streamflow during different
seasons indicates potential inadequacy of the model in effectively representing the complex hydrological processes occurring
during various seasons, including snowmelt and spring runoff. For the study period, rain-on-snow events are identified based
upon the occurrence of precipitation during winter months, along with a decrease in snow depth, coupled with a maximum
temperature that exceeds 0°C. In this experiment, the model can detect melting patterns during these events, but it exhibits
limitations in capturing high and low peaks (Figure 4). As rain-on-snow events during winter produce runoff, the model tends
to interpret these as streamflow. Simulated streamflow is relatively similar to the observed streamflow after the events. For





**Table 3.** All Calibration Experiments with their corresponding parameters and objective function values

| Parameters | Calibration Experiments (2014-2020) | | | | | | |
| --- | --- | --- | --- | --- | --- | --- | --- |
| | DDS | | | PADDS | | | |
| | 1. NSE$_Q$ | 2. RMSE$_{SWE}$ | 3. SPAEF$_{SWE}$ | 4. NSE$_Q$, RMSE$_{SWE}$ | 5. NSE$_Q$, SPAEF$_{SWE}$ | 6. RMSE$_{SWE}$, SPAEF$_{SWE}$ | 7. NSE$_Q$, RMSE$_{SWE}$, SPAEF$_{SWE}$ |
| 1. Base refreezing temperature (mm/d) | -1.049 | -1.138 | -1.879 | -2.174 | -1.737 | -1.782 | -1.896 |
| 2. Temperature threshold for melt-Coniferous (°C) | 0.827 | -4 | 3.374 | 0.695 | 2.817 | 2.483 | 1.806 |
| 3. Temperature threshold for melt-Deciduous (°C) | 0.241 | 4 | -1.797 | -0.838 | -3.391 | -3.116 | -1.778 |
| 4. Temperature threshold for melt-Open (°C) | -1.28 | 4 | -2.224 | -2.767 | -2.987 | -3.913 | 0.542 |
| 5. Melt factor for coniferous forests (mm/d per °C) | 13.591 | 15 | 13.732 | 5.515 | 11.443 | 13.055 | 9.445 |
| 6. Melt factor for deciduous forests (mm/d per °C) | 3.913 | 15 | 11.465 | 10.350 | 6.312 | 7.859 | 10.916 |
| 7. Melt factor for open areas (mm/d per °C) | 9.841 | 15 | 13.395 | 9.891 | 4.929 | 13.950 | 13.340 |
| 8. Multiplication factor for PET | 1.031 | 1.166* | 1.039* | 1.047 | 1.034 | 1.322* | 1.086 |
| 9. Depth of the first soil layer (m.) | 0.003 | 0.003* | 0.003* | 0.003 | 0.003 | 0.003* | 0.001 |
| 10. Depth of the second soil layer (m.) | 0.501 | 0.474* | 0.339* | 0.251 | 0.261 | 0.401* | 0.433 |
| 11. Depth of the third soil layer (m.) | 2.095 | 3.932* | 1.000* | 1.068 | 1.000 | 3.175* | 7.000 |
| *NSE$_Q$* | 0.762 | 0.575 | 0.737 | 0.721 | 0.750 | 0.687 | 0.754 |
| *RMSE$_{SWE}$ (mm)* | 45.35 | 35.74 | 39.38 | 39.09 | 40.53 | 38.89 | 40.15 |
| *SPAEF$_{SWE}$* | 0.192 | 0.072 | 0.232 | 0.091 | 0.229 | 0.244 | 0.216 |
| *KGE$_Q$* | 0.772 | 0.658 | 0.764 | 0.721 | 0.775 | 0.820 | 0.805 |
| **Validation (2001-2013)** | | | | | | | |
| *NSE$_Q$* | 0.735 | 0.595 | 0.726 | 0.719 | 0.749 | 0.679 | 0.741 |
| *KGE$_Q$* | 0.684 | 0.504 | 0.708 | 0.687 | 0.766 | 0.659 | 0.676 |

\* These values are obtained after sequential calibration with NSE



**Table 4.** SPAEF$_{SWE}$ values for each year from each experiment with best value of 0.437 (Exp 1-2017) and worst value of -0.270 (Exp 2-2020)

| Year | Exp- 1 | Exp- 2 | Exp- 3 | Exp- 4 | Exp- 5 | Exp-6 | Exp- 7 |
|------|--------|--------|--------|--------|--------|-------|--------|
| 2015 | 0.262 | 0.175 | 0.240 | -0.071 | 0.232 | 0.231 | 0.137 |
| 2016 | 0.317 | 0.000 | 0.300 | 0.205 | 0.332 | 0.345 | 0.324 |
| 2017 | 0.437 | 0.170 | 0.323 | 0.389 | 0.391 | 0.389 | 0.413 |
| 2018 | 0.051 | 0.080 | 0.138 | 0.093 | 0.062 | 0.130 | 0.081 |
| 2019 | 0.116 | 0.275 | 0.252 | 0.147 | 0.223 | 0.278 | 0.217 |
| 2020 | -0.030 | -0.270 | 0.137 | -0.216 | 0.134 | 0.091 | 0.123 |

the calibration period, values of 0.762 and 0.772 are obtained in this experiment for NSE$_Q$ and KGE$_Q$, respectively. When compared with the base model, the NSE$_Q$ value is improved (from 0.630 to 0.762), which is expected given that NSE$_Q$ is the objective function. The KGE$_Q$ value slightly declined from 0.806 to 0.772, suggesting a slight decrease in hydrological model simulation accuracy, indicating a potential mismatch between observed and simulated hydrographs. The model is validated for the 2001-2013 period, and the NSE$_Q$ value that was obtained is 0.735, thereby indicating good agreement between observed

and simulated streamflow values. The KGE$_Q$ value that was obtained is 0.684, which suggests moderate agreement during the validation period. The spatially averaged RMSE$_{SWE}$ value of SWE for the watershed with respect to SNODAS-SWE is 45.35 mm. Figure 4 indicates that HYDROTEL tends to overestimate SWE compared to SNODAS, except for year 2020, where the model produces a substantial amount of streamflow during winter (Figure 3) as compared to SNODAS data. Either insufficient winter precipitation in the hydrological model or inaccurate temperature data for the year 2020 could be contributing factors

to this issue. Upon comparing precipitation data for 2020 with precipitation that was obtained from meteorological stations, discrepancies were observed in the MSWEP precipitation data, particularly some missing peaks during the winter season. These discrepancies in the precipitation data could potentially contribute to the unusual output that was observed in the study. Through the comparative analysis of SWE data that were collected from the Milan (elevation: 496 m) and Bury (elevation: 340 m) stations, it suggests that the calibrated model exhibits a tendency to closely correspond with the values that were

obtained from Milan, a station that was characterized by a higher elevation. This response suggests the substantial influence of the elevation factor on the model simulations. SPAEF is computed for SWE for the month of March for each year of the calibration period and varies from -0.030 for 2020 to 0.437 for 2017 (Table 4), indicating that the success at simulating the spatial SWE patterns by HYDROTEL is highly variable from year to year.

    In Experiment 2, all snow-related parameters are first calibrated using RMSE$_{SWE}$ with spatially averaged (considering

the SWE values for each RHHUs for whole calibration period then averaging), modelled and SNODAS SWE followed by calibration of the remaining parameters with NSE$_Q$ applied to streamflow. The average RMSE$_{SWE}$ value after calibration is 35.74 mm, which is considerably improved compared with the Standard Calibration experiment. Indeed, Figure 5 effectively shows that simulated SWE more closely matches SNODAS SWE compared to the Standard Experiment. Note that for both experiments, HYDROTEL significantly underestimates snow accumulation for winter 2020. The cause for this discrepancy



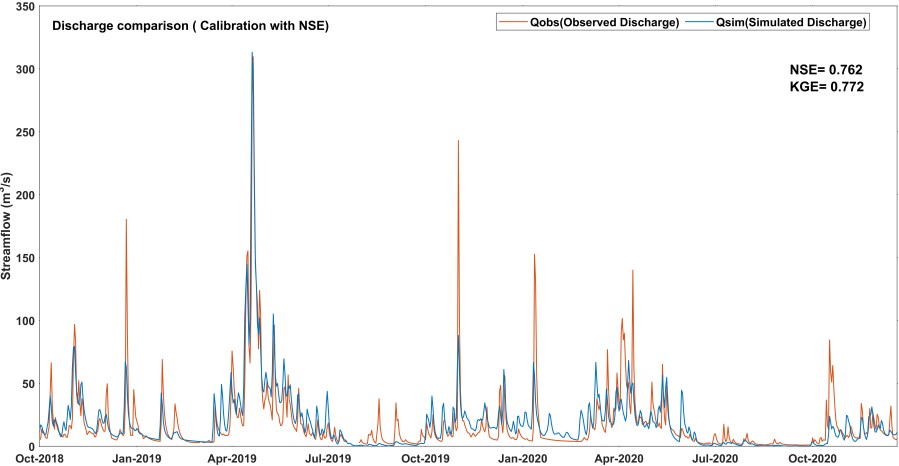

**Figure 3.** Comparison of observed streamflow with simulated streamflow for Experiment 1

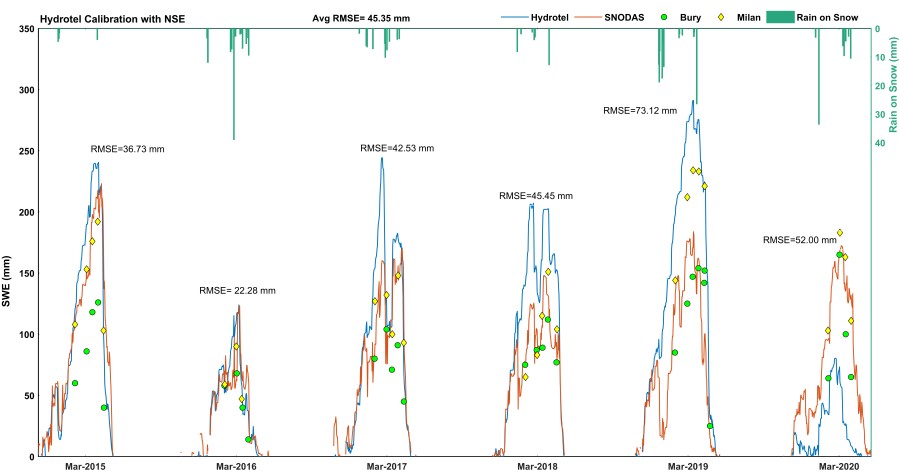

**Figure 4.** Comparison of SNODAS SWE with simulated SWE for Experiment 1 along with station data

remains consistent with the previously discussed reasons. $NSE_Q$ for streamflow after sequential calibration is 0.575 and $KGE_Q$ value is 0.658, which are considerably lower than corresponding values for the Standard Experiment. The model was thus able to improve simulated basin average SWE, but at the expense of a deterioration of the simulated streamflow. During rain on snow events in winter, the model is able to produce the peaks of SWE, but it is unable to capture accurately the associated melting patterns. Moreover, spatial distribution values of SWE varied from -0.270 (for March 2020) to 0.275 (for March 2019)

(Table 4). The average $SPAEF_{SWE}$ value is much lower compared to values obtained with the Standard Experiment. In other words, spatial heterogeneity of the snowpack deteriorates when calibration is performed with the average SWE value.





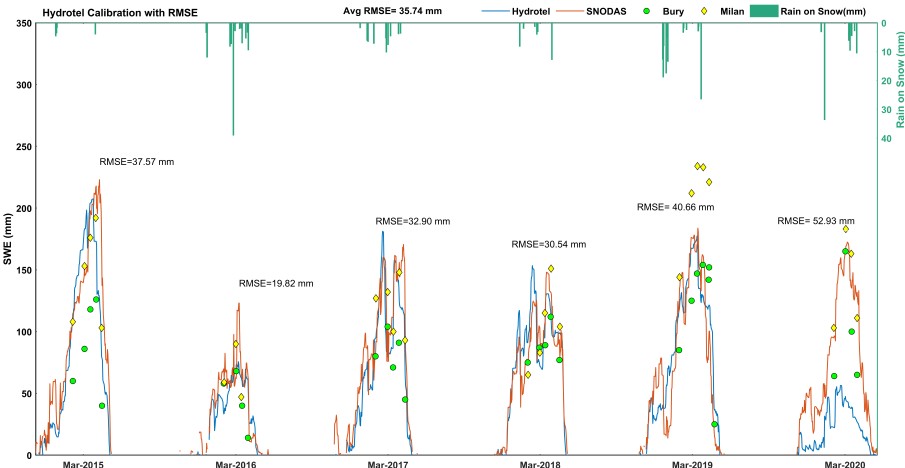

**Figure 5.** Comparison of SNODAS SWE with simulated SWE for Experiment 2 along with station data

Instead of trying to preserve the best temporal dynamics of basin-averaged SWE, Experiment 3 attempts to maintain its spatial distribution at the end of the snow-accumulation season using SNODAS-SWE. This is accomplished by incorporating $\text{SPAEF}_{\text{SWE}}$ as the objective function for SWE for calibrating snow-related parameters, followed $\text{NSE}_{\text{Q}}$ for streamflow to

adjust the remaining parameters. Unsurprisingly, the March $\text{SPAEF}_{\text{SWE}}$ value averaged over years 2015-2020 increased to 0.232, when compared to 0.192 and 0.072 for experiments 1 and 2, respectively (Table 3). Figure 6 depicts the relationship between the spatial distribution of SNODAS and HYDROTEL and the corresponding $\text{SPAEF}_{\text{SWE}}$ value. The results indicate that a greater spatial difference between SNODAS and HYDROTEL leads to a negative $\text{SPAEF}_{\text{SWE}}$ value. Conversely, when the spatial distribution of both datasets is similar, $\text{SPAEF}_{\text{SWE}}$ approaches 1. The figure displays the maximum 0.437 (year-

2017- Experiment 1) and minimum -0.270 (year-2020- Experiment 2) values of $\text{SPAEF}_{\text{SWE}}$ that were obtained during the calibration experiment.

Overall, calibrating HYDROTEL using $\text{SPAEF}_{\text{SWE}}$ helped preserve SWE spatial heterogeneity that was simulated by the model. Also, year-to year variability in $\text{SPAEF}_{\text{SWE}}$ is reduced, as $\text{SPAEF}_{\text{SWE}}$ varied from 0.137 (Mar 2020) to 0.323 (March 2017). A year-to-year comparison of spatial SWE reveals that calibrating the model with $\text{SPAEF}_{\text{SWE}}$ degraded SWE distribu-

tion in some years, e.g., 0.300 (March 2016) and 0.240 (March 2015). This means that spatial integrity of the SWE value is occasionally compromised using the calibration strategy. A detrimental effect of calibrating the model with $\text{SPAEF}_{\text{SWE}}$ is that average SWE value is overestimated by model as compared to observed average value (Figure 7). Correspondingly, average $\text{RMSE}_{\text{SWE}}$ is 39.38 mm, which is higher than the value that is obtained when the model is calibrated using $\text{RMSE}_{\text{SWE}}$ as the objective function. The sequential calibration with $\text{NSE}_{\text{Q}}$ yields a value of 0.737. The $\text{KGE}_{\text{Q}}$ value is 0.764, which is better

than what is achieved in Experiment 2. This suggests that using spatial distribution to calibrate snow parameters apparently provides better results for streamflow than using average SWE value to calibrate snow parameters. Interestingly, both experiments 1 and 3 overestimated spatially averaged SWE, but the sequential calibration strategy provided a good match between simu-





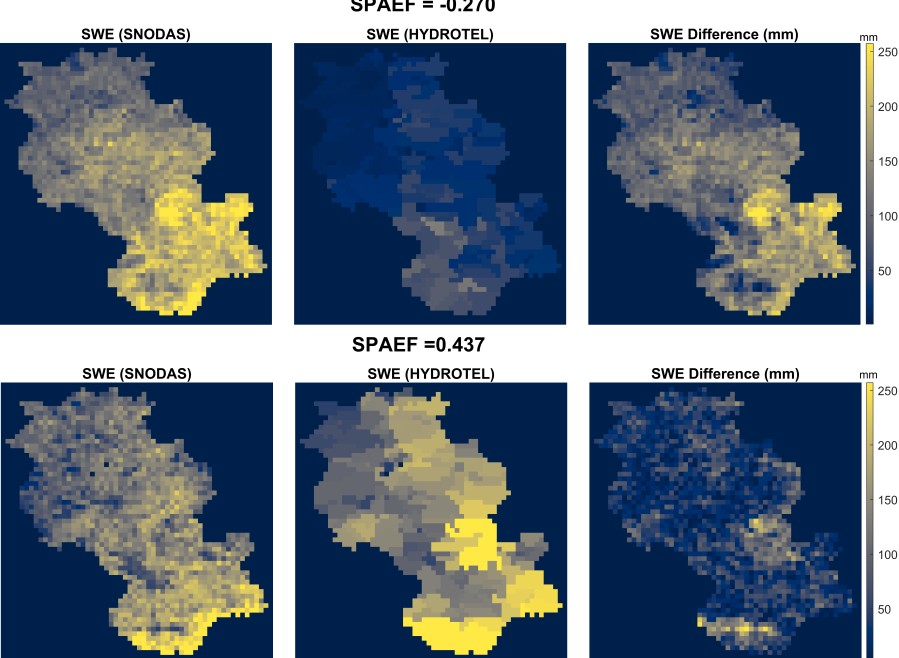

**Figure 6.** Spatially distributed SWE values of SNODAS and HYDROTEL, along with corresponding SWE differences for minimum and maximum SPAEF$_{SWE}$ values for Au Saumon Watershed.

lated versus observed flows, given that NSE$_Q$ and KGE$_Q$ values for experiments 3 and 1 are comparable. From an experimental perspective, it is worth noting that in Experiment 3, the spatial distribution of SWE, i.e., the SPAEF$_{SWE}$ value, exhibits better
improvement when compared to the standard practice. Although the temporal dynamics of spatially averaged SWE are well preserved in Experiment 2, the flow-related model parameters could not be properly calibrated to obtain a good fit between observed and simulated flows. Perhaps this is due to the spatial invariance of these parameters, to the sequential modelling strategy, or both. In order to investigate the latter, experiments were performed in which NSE$_Q$, RMSE$_{SWE}$ or SPAEF$_{SWE}$ are simultaneously optimized with multi-objective calibrations (experiments 4 to 7).

In Experiment 4, NSE$_Q$ for streamflow and RMSE$_{SWE}$ for SWE are optimized together using PADDS. The maximum value of NSE$_Q$ is 0.721, while RMSE$_{SWE}$ is 39.09 mm, which shows improvement compared to the Standard Experiment (Experiment 1) in terms of RMSE$_{SWE}$. Upon comparison with the sequentially calibrated Experiment 2, an improvement was observed in the NSE$_Q$ and SPAEF$_{SWE}$ values, coupled with a decrease in RMSE$_{SWE}$. Surprisingly, the comparison between Experiment 3 and 4 suggests that sequential calibration of the hydrological model using SPAEF$_{SWE}$ results in better model
performance in terms of NSE$_Q$, SPAEF$_{SWE}$, and KGE$_Q$, in contrast to multi-objective calibration with NSE$_Q$ and RMSE$_{SWE}$. SPAEF$_{SWE}$ of SWE varied from -0.216 (for March 2020) to 0.389 (for March 2017), with an average 0.091 for all calibrated years, which is higher than the value that was obtained using RMSE$_{SWE}$ and NSE$_Q$ in a sequential calibration strategy (0.072).





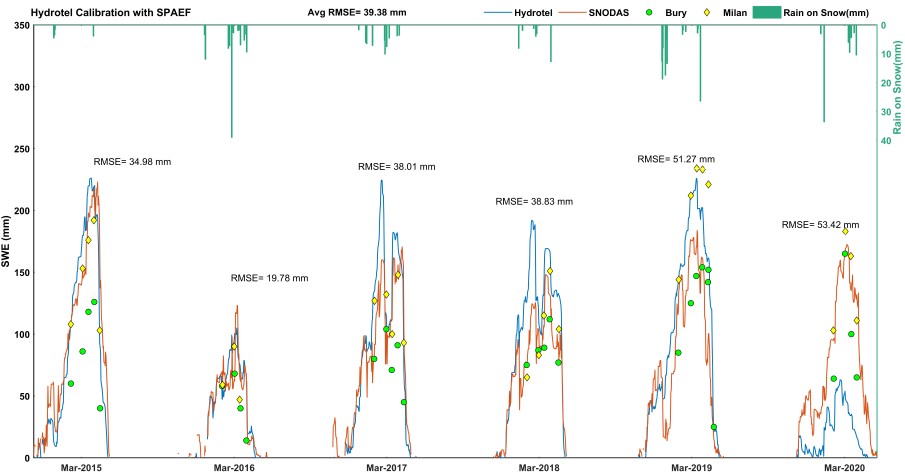

**Figure 7.** Comparison of SNODAS SWE with simulated SWE for Experiment 3 along with station data

In other words, opting for a simultaneous RMSE$_{\text{SWE}}$-NSE$_{\text{Q}}$ calibration improved the spatial SWE distribution compared to a sequential calibration strategy. The value remains below that obtained in Experiment 3 (0.232).

In Experiment 5, NSE$_{\text{Q}}$ is used to optimize streamflow and SPAEF$_{\text{SWE}}$ for spatial SWE together. The optimized solution yields NSE$_{\text{Q}}$ of 0.750 and a KGE$_{\text{Q}}$ of 0.775. Spatial distribution of SWE varied from 0.062 (for March 2018) to 0.391 (for March 2017). The average SPAEF$_{\text{SWE}}$ value that was obtained for all calibrated years is 0.229. When compared with standard calibration (Experiment 1), the spatial distribution of SWE is improved (from 0.192 to 0.229), together with RMSE$_{\text{SWE}}$ (from 45.35 mm to 40.53 mm), while NSE$_{\text{Q}}$ of Experiment 5 is comparable with standard calibration with an improved KGE$_{\text{Q}}$ value.

Upon comparing experiments 5 and 3, where SPAEF$_{\text{SWE}}$ is calibrated sequentially following NSE$_{\text{Q}}$, a slight improvement in NSE$_{\text{Q}}$ is noted. Yet, SPAEF$_{\text{SWE}}$ and RMSE$_{\text{SWE}}$ appear to be compromised. Here, sequential calibration using SPAEF$_{\text{SWE}}$ results in superior performance for RMSE$_{\text{SWE}}$ and SPAEF$_{\text{SWE}}$, whereas multi-objective calibration jointly yields a better performance measure for NSE$_{\text{Q}}$. Comparing the results to Experiment 4, where NSE$_{\text{Q}}$ is calibrated with RMSE$_{\text{SWE}}$, it is observed that NSE$_{\text{Q}}$, KGE$_{\text{Q}}$ and SPAEF$_{\text{SWE}}$ are improved in Experiment 5, while RMSE$_{\text{SWE}}$ results are comparable. This

again suggests that using SPAEF$_{\text{SWE}}$ to calibrate spatially distributed SWE is more advantageous than using RMSE$_{\text{SWE}}$ to calibrate spatially averaged SWE when employing multi-objective functions with NSE$_{\text{Q}}$.

    In Experiment 6, both RMSE$_{\text{SWE}}$ for spatially averaged SWE and SPAEF$_{\text{SWE}}$ for spatial distributed SWE are first optimized simultaneously using PADDS to calibrate HYDROTEL's snow-related parameters, followed by maximizing NSE$_{\text{Q}}$ for streamflow to calibrate the remaining flow-related parameters. After calibration, the best value for RMSE$_{\text{SWE}}$ is 38.89 mm,

while SPAEF$_{\text{SWE}}$ values ranged from 0.091 (for March 2020) to 0.389 (for March 2017), with an average value of 0.244. Sequential calibration with NSE$_{\text{Q}}$ provided NSE$_{\text{Q}}$ value of 0.687 for streamflow and a KGE$_{\text{Q}}$ value of 0.820. In comparison to experiments 4 and 5, NSE$_{\text{Q}}$ has decreased substantially while the KGE$_{\text{Q}}$ value has increased substantially. This suggests that model performance has improved in terms of capturing the overall pattern of the observed data, although the accuracy in fitting





individual data points may have declined slightly. In Experiment 6, a significant improvement is noted for SPAEF$_{\text{SWE}}$, with a

slight decrease in RMSE$_{\text{SWE}}$ compared to Experiment 4. This suggests that by sequentially calibrating both SPAEF$_{\text{SWE}}$ and RMSE$_{\text{SWE}}$ followed by NSE$_{\text{Q}}$, the model is able to capture the spatial distribution of both SWE and streamflow. Yet, it should be noted that model's fitness to individual data points might not be captured accurately. In Experiment 6, slight improvement in SPAEF$_{\text{SWE}}$ was noted, with a slight decrease in RMSE$_{\text{SWE}}$, but a significant reduction in NSE$_{\text{Q}}$ as compared to Experiment 5. This implies that calibrating SPAEF$_{\text{SWE}}$ and NSE$_{\text{Q}}$ together is a better approach than sequential calibration of SPAEF$_{\text{SWE}}$

and RMSE$_{\text{SWE}}$, followed by NSE$_{\text{Q}}$.

As the last step in calibration trial, all objective functions were calibrated together using the PADDS algorithm. NSE$_{\text{Q}}$ after optimization is 0.754, KGE$_{\text{Q}}$ is 0.805, and RMSE$_{\text{SWE}}$ is 40.15 mm. Spatial distribution of SWE varied from 0.081 (for March 2018) to 0.413 (for March 2017). The average SPAEF$_{\text{SWE}}$ value for all calibrated years is 0.216. When compared to the standard experiment, Experiment 7 outperforms in terms of RMSE$_{\text{SWE}}$, SPAEF$_{\text{SWE}}$ and KGE$_{\text{Q}}$, while NSE$_{\text{Q}}$ remains

comparable in both cases. Among other experiments, Experiment 7 shows better performance when compared to experiments 2 and 4, while results from experiment, 3 and 5 are comparable to Experiment 7. By comparing Experiment 6 where NSE$_{\text{Q}}$ is sequentially calibrated with RMSE$_{\text{SWE}}$ and SPAEF$_{\text{SWE}}$, and Experiment 7 where all three functions are calibrated together, we conclude that calibrating together provides better results for NSE$_{\text{Q}}$ and comparable results for other objective functions.

In comparing all calibration strategies during validation, NSE$_{\text{Q}}$ values for the experiments could be ordered: 5, 7, 1, 3, 4, 6,

and 2. KGE values > 0.75 are generally considered to be indicative of good model performance, as noted in previous studies (Towner et al., 2019). Upon analyzing the results in calibration experiments, most are found to have KGE values greater than 0.75 for the calibration period; the exceptions are the second (calibration with RMSE$_{\text{SWE}}$ and sequential NSE$_{\text{Q}}$) and fourth (calibration with RMSE$_{\text{SWE}}$ and NSE$_{\text{Q}}$ simultaneously) experiments. This suggests that calibration in these experiments is satisfactory, and the model is expected to perform well. The validation results from the years 2001-2013 were analyzed for

the best model performance with respect to KGE$_{\text{Q}}$. Experiment 5 had the highest KGE$_{\text{Q}}$ (> 0.75), indicating the best model performance. Experiment 3 followed closely behind, while experiments 1, 4 and 7 produced nearly identical results. In contrast, experiments 2 and 6 had poor performance in terms of KGE$_{\text{Q}}$. A noticeable feature of SPAEF$_{\text{SWE}}$er is the amount of time that is required for calibration, together with the number of iterations to reach the best value. In this study, iterations were set at same number to maintain comparable scenarios, while the duration of spatial calibration was twice as long as the remaining

experiments.

## 5 Discussion

Analysis of parameter variations following calibration revealed consistent values for the base refreezing temperature, the PET parameter, and depth of the first two soil layers across multiple calibrations. Yet, variation was observed in the temperature thresholds for melt, melt factor, and thickness of the third soil layer among experiments, particularly for Experiment 2. In the

initial phase of calibration experiments, several trial and error runs were conducted to determine parameter boundaries, while simultaneously reviewing relevant literature, which enhanced understanding and accuracy through comprehensive parameter



exploration. Experiment 2, which used $RMSE_{SWE}$ and $NSE_Q$ sequentially, consistently reached the parameter bounds for temperature thresholds and melt factors. Depending upon the land use, temperature threshold values also show opposite values, i.e., -4 °C for conifers versus 4 °C for deciduous and open areas. This means that there are areas with a lot of snow and others

with very little snow in the watershed based on land use, which does not represent the accurate spatial distribution of snow for this watershed. The watershed exhibits a significant predominance of coniferous vegetation, leading to a lower temperature threshold in Experiment 2 compared to the other experiments. Experiment 1 overestimates SWE compared to SNODAS. By decreasing the temperature at which melting begins for a large portion of the watershed, Experiment 2 decreases the overall quantity of snow to levels closer to SNODAS values. Yet, the spatial distribution of snow is not respected. Therefore, it is not

recommended that parameters using $RMSE_{SWE}$ and $NSE_Q$ be calibrated sequentially.

As the research objective, this study evaluated the practicality of using raw SNODAS data for hydrological model calibration. Given its specific focus, bias correction of the SNODAS data was not within the scope of the study. As a result, raw SNODAS data were employed for analysis of SWE, and both RMSE and SPAEF were utilized as objective functions to calibrate SWE in the model. In Experiment 2, $RMSE_{SWE}$ can drive the parameters to extreme values, given that it treats all data points equally

irrespective of their location in the distribution. If there are extreme values in the observed data, the model can be calibrated to fit those values, even if they do not represent the overall distribution. This can lead to poor model performance when applied to new data or different conditions. Furthermore, if there is bias in the observed data, it will result in high RMSE values. In such situations, SPAEF may be a more reasonable option to achieve good calibration of SWE when bias correction of data is not feasible.

The calibration of streamflow alone may not be sufficient for accurately simulating the hydrological processes in a watershed, given that it is affected by various factors, such as snow accumulation, snowmelt, evapotranspiration, infiltration and soil moisture. Therefore, considering the calibration of SWE is crucial in watersheds that receive substantial snowfall, such as the Au Saumon River Watershed, where snow accumulation and melt are major contributors to the streamflow. Calibrating only streamflow may not capture the snow accumulation and melt dynamics and can lead to inaccurate simulation results,

which in turn can affect water resources management decisions. In their study, Nemri and Kinnard (2020) investigated the integration of snow observations into improving calibration of hydrological models in forested catchments. They found that employing multi-objective calibration, which incorporates both streamflow and SWE (both calibrated using NSE), resulted in SWE simulations that were comparable to a separate calibration method with a small decline in streamflow simulations. The researchers emphasized the importance of considering the spatial distribution of the data when calibrating the models.

Our study's focus aimed to incorporate snow spatial information for hydrological model calibration. A novel spatial efficiency metric called SPAEF was utilized in conjunction with other objective functions, i.e., RMSE and NSE. NSE was employed as an objective function for streamflow. It enables direct evaluation of model performance against the inherent benchmark of NSE = 0, which corresponds to the mean flow. Alongside other metrics, KGE was computed for both calibration and validation. Knoben et al. (2019) suggested that KGE values falling within the range of -0.41 < KGE ≤ 1 can be considered

reasonable for hydrological modelling. This indicated a satisfactory representation of the observed data, taking into account the limitations and uncertainties that were associated with the model and data. Consequently, KGE was utilized as a performance





metric to compare validation results, with Table 3 summarizing the corresponding parameter values for each calibration experiment. Based upon the comparison of validation results using $KGE_Q$, it is evident that incorporating spatial calibration with $SPAEF_{SWE}$ in conjunction with $NSE_Q$ (in both simultaneous and sequential calibrations) yields better outcomes compared to

utilizing $RMSE_{SWE}$ as one of the objective functions or using $NSE_Q$ as only objective function.

Looking at first three experiments, it can be inferred that the sequential calibration of NSE following the calibration of spatially distributed SWE with $SPAEF_{SWE}$ yields outcomes that exhibit better acceptability as the overall model performance is enhanced. The reason for this is that calibrating SWE captures the spatial variability of the snowpack, which is a crucial factor in hydrological processes. Calibrating $NSE_Q$ subsequently ensures that the model can capture temporal variation of

the flow. Therefore, the sequential calibration approach leads to better results that are acceptable in terms of overall model performance. When comparing results of calibration with $RMSE_{SWE}$ and calibration with $SPAEF_{SWE}$, followed by sequential $NSE_Q$ calibration, it is evident that $SPAEF_{SWE}$ yields better results than $RMSE_{SWE}$. The distribution of snow is not uniform everywhere; therefore, spatially distributed SWE calibration captures the heterogeneity of the snow distribution within the basin, whereas spatially averaged snow calibration assumes that snow is uniformly distributed throughout the basin, which

is not always the case in mountainous terrain where snow can accumulate in complex patterns. Thus, spatially distributed SWE calibration provides a more accurate estimate of the actual snow distribution in the basin, which leads to better model performance in predicting streamflow.

In a comparative analysis of hydrological model calibration procedures, (Tuo et al., 2018) examined the effectiveness of different calibration approaches. Their study focused upon the multi-objective calibration method, specifically incorporating

the optimization of sub-basin average snow water equivalent (SWE) and streamflow. The results demonstrated that this multi-objective approach outperformed single-objective procedures in accurately simulating snow dynamics, which aligns with our study. Building upon these findings, our study extended the calibration process by further incorporating both spatially averaged and spatially distributed data for SWE. Notably, our results highlighted the superiority of calibrating the model using spatially distributed information rather than relying solely upon average information. Considering the spatial distribution of SWE data

leads to improved model performance and more accurate simulations.

Focusing upon multi-objective calibrations, experiments 4 and 5 also backs up the aforementioned argument that using $SPAEF_{SWE}$ with $NSE_Q$ yields better results than using $RMSE_{SWE}$ with $NSE_Q$. Based upon the comparisons made between experiments 2 vs. 4, 3 vs. 5, and 6 vs. 7, it is evident that calibrating the objective functions simultaneously yields superior model performance compared to the sequential calibration of the objectives. Specifically, experiments 4, 5 and 7, which employ

the simultaneous calibration of objective functions that were considered, exhibit improved model performance when compared to experiments 2, 3 and 6, which adopt a sequential calibration approach. The study that was conducted by Finger et al. (2015) showcased the benefits of calibrating a hydrological model using multiple data sets, thereby leading to improved estimation of runoff contribution. This finding is consistent with the current study, which highlights calibrating both SWE and streamflow as yielding superior results. From our study, it can be concluded that simultaneous calibration of objective functions is a superior

approach to sequential calibration, given that sequential calibration can lead to overfitting of the model to the specific objective function being calibrated. In turn, this can result in poor model performance when evaluating other objective functions. Fur-





thermore, sequential calibration may result in a trade-off between objective functions, which may not be optimal for overall model performance. When all objective functions are calibrated simultaneously, it allows for a more balanced calibration and can provide better overall model performance. It also helps avoiding overfitting to any single objective function and provides a more comprehensive understanding of the model's behaviour.

## 6 Conclusions

Hydrological models are subject to continuous development, which has led to increased complexity. These models are not merely tools for estimating runoff; rather, they encompass complex processes that involve state variables contributing to the generation of runoff. The satellite input data, which is used to drive these models, is available at high temporal and spatial resolutions. Integration of comprehensive spatial data that are acquired from remote sensing platforms offers tremendous opportunities for further advancements in hydrological modelling.

This article analyzes different calibration experiments of the HYDROTEL distributed hydrological model for the Au Saumon Watershed. HYDROTEL includes modules that permit high-resolution discretization of the basin, river streams, lake inflow, river flow, and gridded observed meteorological data, making it a suitable model for the calibration experiment. The key aspect of this calibration experiment is the incorporation of the spatial efficiency metric SPAEF as an objective function. The study explored this newly developed spatial distribution metric in the calibration and validation of distributed hydrological models and compared results with previously used calibration strategies. SPAEF has been used previously with evapotranspiration in various studies, but this study introduces SPAEF with SWE for the first time. The comparison of different calibration strategies on the Au Saumon Watershed highlights these important findings.

- Calibrating only streamflow is not ideal for any hydrological model. It is recommended that snow parameters such as snow water equivalent (SWE) also be calibrated, especially in areas where snow accumulation can be spatially heterogeneous.

- Sequential calibration of objective functions (e.g., calibrating using NSE after calibrating with SPAEF) may not always result in better model performance compared to calibrating all objective functions simultaneously, especially when considering multiple objective functions. Sequential calibration of objective functions is not recommended, as it may result in sub-optimal model performance.

- Spatially distributed SWE calibration is preferred over spatially averaged calibration, given that the former captures heterogeneity of snow distribution in different land covers and provides more accurate estimates of SWE across the basin.

- Raw SNODAS data has the potential for enhancing the model's accuracy and reliability by incorporating the spatial variability of snow distribution.



The present experiments demonstrate that although researchers tend to focus upon obtaining decent model output by optimizing a single objective function, this approach may not provide entirely reliable results. Therefore, using multiple objective functions to optimize different processes simultaneously can lead to better results. In this study, the importance of incorporating the spatial calibration metric SPAEF is highlighted. Spatial calibration of snow parameters provides better results when compared to averaging the parameters. To further understand the spatial metric, it is necessary to investigate spatial variability and SPAEF by applying and comparing it to other catchments or models. Calibrating a distributed model and increasing its spatial predictability requires more than just an appropriate spatial performance indicator. It necessitates the use of a flexible model structure and parameterization in conjunction with other metrics to enable simulated patterns to be modified meaningfully. Achieving this requires reliable geographic measurements at an appropriate scale, thorough assessment of catchment morphology, and high-quality forcing data.

Based upon our findings, it is evident that spatial calibration of a distributed hydrological model, HYDROTEL, yields satisfactory results and enhances its robustness and coherence with other hydrological processes. Our study aims to encourage the modelling community to reconsider their methodologies by focusing upon relevant metrics that emphasize spatial patterns characterizing hydrological processes during calibration or validation studies. The upcoming Terrestrial Snow Mass Mission (TSMM) satellite mission seeks to offer high-resolution and spatially distributed information on snow water equivalent (SWE). Consequently, to optimize hydrological model performance, calibration procedures that account for both conventional streamflow and spatial SWE should be considered.

*Code and data availability.* The SPAEF code that is used in this study for spatial performance metrics is available at https://github.com/cuneyd/spaef. The data that are used in this study are openly available for download from the respective websites: https://cds.climate.copernicus.eu,https://www.gloh2o.org/mswep/, and https://nsidc.org/data/g02158/versions/1.

*Author contributions.* Dipti Tiwari (DT) ,Mélanie Trudel (MT) and Robert Leconte (RL) designed the study. DT collected data, performed the calibration and validation study and drafted the manuscript. MT and RL assisted in structuring the article. The editing of the article, as well as the discussion and interpretation of the findings, were all co-authored.

*Competing interests.* The authors declare that they have no conflicts of interest.

*Acknowledgements.* The authors would like to thank our industrial partners Hydro-Québec, Brookfield and Ville de Sherbrooke. Funding for this work was provided by the Natural Sciences and Engineering Research Council of Canada (NSERC).





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
