# Peer review of "On optimization of calibrations of a distributed hydrological model with spatially distributed information on snow"

_Hydrology and Earth System Sciences, 2023_

## Referee Comment (RC2)

**Comment on hess-2023-143**

This study by *Tiwari et al* presents a novel utilization of a spatial metric SPAEF for calibrating and investigating the appropriateness of raw SNODAS snow water equivalent information in distributed hydrological modeling. The manuscript compares different calibration approaches including sequential, and multi-objective experiments using SWE and discharge, and a traditional single variable (discharge) based calibration, using different global optimization routines. The study is based in a snow-dominated catchment in Canada. The results indicate a better simulation of the spatial distribution of the SWE when SPAEF is used in calibration, with a more robust hydrological prediction.

I read the paper with interest given the similarity with my recent works. However, I have to say that this manuscript, in its current form, must undergo major revision to enhance the clarity for readers and for publication. The justification for this claim is based on the following review comments from my side.

**General comments:**

1. The introduction section is not coherent and should be improved. I agree with Reviewer 1 that a major streamlining of the introductory section must be done. For instance, the description of SNODAS data can be brought together and shortened. Similarly with the problems with hydrological models while using single discharge variable for calibration, leading to parameter compensations can be merged. Please avoid repeating statements. A suggestion would be to streamline the introduction section with explanations about the objective functions and optimization routine briefly explained in different sections. TSMM justification can be provided in the introduction section or conclusion, to avoid confusion for the results.
2. Consider putting a short novelty of the research in the abstract as well and shorten it with specific information from your research.
3. The novelty of the study should be explicitly highlighted in the introductory section along with major findings. Multi-objective calibration is not a novel approach and has been identified to provide more reliable snow estimations with better or similar hydrological performance in different studies. However, the use of SPAEF for calibrating raw SNODAS data, as a novelty, should be highlighted comparing with past research employing similar approach with SNODAS or other data. The SPAEF metric used in other modeling approaches (Line 527) can be pulled to the introduction section.
4. The usage of 'parameters' and 'variables' should be uniform as it can impart different meanings to the readers.
5. Please make a clear distinction between the spatial unit used for calibration. Is it RHHU or pixel based? If pixels, how is the spatial information (esp. SNODAS data) aggregated at RHHUs?
6. How is the discharge extrapolated to the basin outlet?
7. The reasons behind calibrating the model solely against SWE distribution for March (peak accumulation) should be elaborated. Why not take the whole snow-season? How sensitive are the results when the calibration is done during the onset or the melt phases of the snow season? Since the study focuses on how raw SWE data can be used to constrain a hydrological model calibration, these sensitivity tests would further add value for other researchers and practitioners, for operational use.
8. I agree with Reviewer 1 on the explanation of RMSE and NSE in the objective function section. This can be reduced. Also consider explaining why KGE specifically was used for validation.

9. Given the coarser resolutions of the input drivers, how is the elevation dependent temperature and precipitation trends accounted for by Thiessen polygon method? This can have a detrimental effect on the simulation of snow accumulation and melt processes. Also, regarding the base model parameters used for comparison of model performance, did the cited research work with similar global model drivers? Consider giving a short explanation to formulate a sound basis for comparison.
10. Avoid reexplaining the calibration strategies, which makes the whole section longer.
11. Streamline the discussion section focusing on the results and corresponding relevant literature. Please avoid repetitions within the section and with the introduction. A short discussion on the uncertainties related to input data and SNODAS as compared to station observations would be better. Additional information on snow improving the hydrological understanding is not novel. Please focus on how your approach better represents the snow processes as well as parameter identifiability.

12. Please specify the base conditions used for calibration (i.e choice of input data, model and the calibration variable) for strong conclusions like Concluding remarks #2, considering the study was done in only one catchment.
13. How does this research differ from those who use bias corrected SNODAS information, particularly in terms of capturing the spatial distribution of snow cover and discharge simulation?

**Specific comments:**

Line 2: Please make a clear and uniform distinction between snow 'parameters' and 'variables'.

Line 4: Full form of SWE at the first instance.

Line 8: Remove 'approach'. 'calibration' should suffice.

Line 9: Full form of SNODAS at the first instance.

Line 14: Which "model performance"? Consider adding SWE and discharge simulation.

Line 5: 'hydrological events'

Line 56: Remove 'which'.

Line 66: Add references here.

Lines 101 - 102: Consider rephrasing the sentence 'Ensuring….. streamflow forecasts'.

Line 136: 'spatialized'?

Line 150: Consider 'higher elevations' instead of 'hilly places'.

Line 175: What does 'other modules' mean? Consider rephrasing.

Line 178: 'discharge is simulated' not 'model'

Line 202: Please check '2015-2020'.

Lines 214-215: Not necessarily. Add references or please rephrase.

Line 231: flow simulated at corresponding RHHU or at the outlet?

Lines 268 – 269: Please refer to general comments 5 on making a distinction between grid or RHHU. Better to make it clear beforehand in model description section.

Line 280: 'run with some random parameter values' instead of 'ran with some random value'.

Line 306: It would be interesting to look at the SPAEF values for the base model as well.

Line 318: Describe figure 4 in a better way than this. The current statement is confusing. Also please explain the significance of adding the 'rain-on-snow' description in this section.

Table 4: Please add base model performance as well.

Lines 327-328: Consider rephrasing this sentence about Figure 4.

Lines 336-338: Consider rephrasing the sentence 'SPAEF….. year to year'. It will be interesting to see the SPAEF values for the onset and depletion phases of the snow season as well.

Figures 3, 4, 5 and 7: Bigger font size please and also check the uniformity of the titles of these figures as well (esp. on top left). Consider plotting an average SWE line (simulated and observed) in these figures.

Lines 356 – 361: "relationship" or "comparison" ? Consider moving this section earlier or please put the experiment information in the caption in Figure 6.

Line 395: Please check if the calibration is done following 'NSE' based calibration.

Line 415: "under the considered model setting".

Line 432: Check $SPAEF_{SWE}$

Lines 431-432: Explain more on this or remove this sentence about computational efficiency altogether.

Line 516: This sentence is confusing. Please rephrase.

Line 518: 'simulation of complex processes' rather than 'complex processes'

Line 519: What is the high resolution defined here?

Line 530: 'Snow variables' instead of 'snow parameters'

Overall, this research has some interesting findings on how raw SNODAS data can complement the existing hydrological modeling techniques, especially with a novel use of a SPAEF metric in a multi-objective calibration setting. However, the structure of the manuscript must be substantially revised for further acceptance. I would thus recommend further consideration of the manuscript after major revision.

---

## Author Response (AR1)

**RESPONSE TO REVIEWER 1:**

We thank the reviewer-1 for the thorough review and comments. We have now addressed all the points raised by the reviewer. Below are our detailed responses to the reviewer's comments, along with the respective changes that we will make in the manuscript. We hereby address them individually. In this document we indicate the Reviewer's comments in *italic red*, while text that was changed in the paper in blue. The line numbers mentioned here is in the updated manuscript.

*The manuscript by Tiwari et al. presents the use of a novel metric, SPAEF, for estimating the spatial variability of hydrological variables during hydrological model calibration. Various calibration strategies, including stepwise calibration and traditional global optimization methods, were employed. To enhance clarity and conciseness, the abstract and introduction should be streamlined to ensure that readers can easily grasp the take-home messages from the study. Some introductory content may be relocated to other sections. Furthermore, it is essential to clearly and succinctly illustrate the novelty of this study.*

In response to the reviewer's comments on enhancing clarity, conciseness, and emphasizing the novelty of the study, we have undertaken the following revisions to the manuscript:

- Refined the abstract to summarize the key findings and contributions briefly.
- Streamlined the introduction by eliminating unnecessary details and relocating introductory content for better alignment to ensure that the key insights conveyed by the study is effectively communicated to the readers.
- Articulated the unique aspects of the study, particularly the innovative use of SPAEF and the impact of different calibration strategies.

*Lines 3-6: These two sentences appear somewhat unrelated to the main focus of this study. Consider removing them from the abstract.*

Removed the lines 3-6 in updated manuscript.

*Line 8: What is SNODAS? It might not be well known (which may not be true). Please provide a complete description of this acronym.*

Provided complete description of the acronym for SNODAS (Snow Data Assimilation System)

*Lines 9-11: This information may not constitute the primary takeaway from the paper and may not be necessary.*

We agree with the reviewer, as this information does not contribute to the primary takeaway ,we removed it from abstract.

*Line 21: Consider adding a sentence at the end of the abstract to explicitly emphasize the novelty of this study.*

Added a line emphasizing the novelty of the study "The novelty of this study is the implementation of SPAEF with respect to spatially distributed SWE for calibrating a distributed hydrological model. Lines 17-18

*Line 35: Consider specifying "during the snow-melt season."*

Specified "snow-melt season, typically from March to May" Lines 33

*Line 52: Please provide specific details regarding the fine resolution.*

Added specific details of resolution "GlobSnow resolution (25 km * 25 km)- Lines 47, SNODAS (SNOw Data Assimilation System) resolution (1 km* 1 km)". Lines 50

*Lines 54-58: Consider relocating these sentences to the discussion or conclusion section to avoid confusion among readers regarding the use of TSMM SWE.*

We agree with the reviewer, moved the lines to conclusion for better clarity in updated manuscript.

*Line 94: Consider using "variable" instead of "parameter" to clearly distinguish between model parameters and hydrological variables.*

Replaced parameters with variables for better clarity between model parameters and hydrological variables.

*Lines 115-141: This paragraph could be relocated to the methodology section. Additionally, in the description of the optimization algorithms, please clarify the number of repeated experiments conducted to obtain parameter values.*

Moved the paragraph to methodology and added "A total of 1000 iterations conducted for both DDS and PADDS to optimize parameter values." Lines 211-212

*Line 115: The final paragraph of the introduction should emphasize the novelty of this study, potentially by comparing it to current methodologies or existing approaches for optimizing model parameters.*

"The primary objective in this study is to introduce spatial calibration with SWE data using newly developed metric SPAEF for the calibration of the HYDROTEL hydrological model. We applied SPAEF in combination with other traditionally used objective functions. We conducted seven distinct calibration experiments, each employing a unique combination of objective functions. This allowed us to assess the trade-offs and robustness of these various calibration scenarios by evaluating their performance in terms of both streamflow and spatial SWE patterns. Notably, while SPAEF has been previously applied in studies involving evapotranspiration (Demirel et al., 2018) and soil moisture (Eini et al., 2023), this study uses SPAEF with SWE for the first time." Lines 109-115

*Line 164: Add "the variation of" before SWE for clarity.*

Corrected in manuscript.

*Line 169: Please clarify the time step of the model employed in this study.*

simulated on a daily basis (opted for this study). Lines 142

*Line 177: Please explain the selection of the Thiessen polygon method. Does the elevation change the precipitation (lapse rate)?*

The Thiessen polygon method is selected to interpolate the meteorological data spatially, the values recorded at the station nearest to a specific cell is allocated to that respective cell. When Thiessen polygon is applied on a gridded data, the polygons generated outlines the area where a specific grid cell is the closest .Therefore, the Thiessen polygons align with the precipitation grid and here we can see that the application of a Thiessen polygon is equivalent to the grid.

"with vertical precipitation gradient of 1mm/100m and vertical temperature gradient of -1°C/100m." Lines 151-152

[Figure]

*Line 205: Specify the nature of the data used for March. Is it the mean value of March or daily values?*

"average SWE of March" Lines 181

*Line 223: As mentioned earlier, please specify the time step of the streamflow data.*

"The streamflow data is available in a daily basis" Lines 175

*Line 226: The sentence regarding NSE and similar sentences describing RMSE, KGE, etc., may not be necessary. These details are more suitable for a project report than a science paper.*

Removed sentences that were not necessary for the study.

*Lines 266-269: Further elaborate on the rationale for selecting March for SPAEF analysis to avoid appearing arbitrary. Consider testing SWE throughout the year or in other months using a similar approach for model calibration and provide this analysis in the Supplementary material.*

Thank you for this comment. In response to your comments, I would like to clarify the rationale behind our selection of March as the month for SPAEF analysis in our study. March was chosen as it is the month with the maximum SWE. Our aim was to utilize the maximum SWE information available during this period. However, we acknowledge that March, despite having the maximum SWE, also coincides with the snow melting period, which could potentially affect the calibration of our analysis. We conducted additional analyses using data from January and February. The results showed that SPAEF performs well with data from both these months. We believe that further research is necessary, with different watersheds and periods used to compute SPAEF, to understand the performance of SPAEF more accurately. The detailed results of these additional calibration can be found in the supplementary material, providing a comprehensive view of the model's performance.

| | Calibrated with respect to SPAEF_March & NSE | Calibrated with respect to SPAEF_February & NSE | Calibrated with respect to SPAEF_January & NSE |
|---|---|---|---|
| NSE | 0.737 | 0.739 | 0.733 |
| KGE | 0.764 | 0.771 | 0.840 |
| RMSE Spatial | 39.38 | 51.90 | 50.23 |
| SPAEF wrt SNODAS Jan | 0.01 | 0.077 | 0.101 |
| SPAEF wrt SNODAS Feb | 0.157 | 0.201 | 0.181 |
| SPAEF wrt SNODAS March | 0.232 | 0.197 | 0.167 |

We added supplementary material with results when calibrated with other months (January and February) and the updated rational in the manuscript.

 "For this study, March was selected for SPAEF calibration as it is the month with the highest SWE. Our objective was to leverage the maximum SWE information available during this period. However, we recognize that March, despite having the highest SWE, also overlaps with the snow melting period, which could potentially influence the calibration of our analysis. We performed additional analyses using data from January and February, and the results demonstrated that SPAEF performs well with data from both these months. We believe that further research is necessary, with different watersheds and periods used to compute SPAEF, to more accurately understand SPAEF's performance during the onset of snow accumulation and the snowmelt period. The detailed results of these additional calibrations can be found in the supplementary material, providing a comprehensive view of the model's performance."
Lines 516-523

*Line 275: Consider placing the KGE metric for model validation in a separate section, distinct from the objective functions used for optimization.*

A separate subsection is introduced " Other metric used in this study: One other metric KGE ( Kling–Gupta efficiency) is computed for all the calibration experiments. It has

been used to assess overall model performance for the various calibration scenarios that were investigated in this study. Lines 269-271

*Lines 457-459: The support for this result appears insufficient. Please provide additional information or clarification.*

The explanation is added "The sensitivity of RMSE to outliers is a common concern while using it in calibration. Outliers can significantly impact RMSE calculations, and their likelihood of occurrence aligns with the normal distribution that underlies RMSE (Chai and Draxler, 2014). When model biases are pronounced, it may be necessary to address these systematic errors before calculating RMSE. However, the bias insensitivity of SPAEF offers a valuable solution to this challenge (Koch et al., 2018). SPAEF mitigates the impact of uncertainties in observations, providing a more robust and stable approach to model calibration and evaluation. Lines 463-468

*Line 530: When drawing the conclusion, be specific and careful about specifying the type of hydrological model and the situations in which this conclusion holds true.*

Corrected the conclusion with "distributed hydrological model" Lines 539

*Figures: The figure axes and the labels should be more obvious. Currently, they are too small.*

Updated the figure fonts.

*Overall, I think the paper exhibits novelty, especially in introducing a new objective function for SWE in model parameter calibration. However, the writing style of the scientific paper can be further improved. I would recommend accepting this paper after moderate revisions.*

Chai, T. and Draxler, R. R.: Root mean square error (RMSE) or mean absolute error (MAE)? – Arguments against avoiding RMSE in the literature, Geoscientific Model Development, 7, 1247–1250, https://doi.org/10.5194/gmd-7-1247-2014, 2014.

Demirel, M. C., Mai, J., Mendiguren, G., Koch, J., Samaniego, L., and Stisen, S.: Combining satellite data and appropriate objective functions for improved spatial pattern performance of a distributed hydrologic model, Hydrology and Earth System Sciences, 22, 1299–1315, 2018.

Eini, M. R., Massari, C., and Piniewski, M.: Satellite-based soil moisture enhances the reliability of agro-hydrological modeling in large transboundary river basins, Science of The Total Environment, 873, 162396, https://doi.org/10.1016/j.scitotenv.2023.162396, 2023.

Koch, J., Demirel, M. C., and Stisen, S.: The SPAtial EFficiency metric (SPAEF): Multiple-component evaluation of spatial patterns for optimization of hydrological models, Geoscientific Model Development, 11, 1873–1886, 2018.

**RESPONSE TO REVIEWER 2:**

We thank the reviewer-2 for the thorough review and comments. We have now addressed all the points raised by the reviewer. Below are our detailed responses to the reviewer's comments, along with the respective changes that we will make in the manuscript. We hereby address them individually. In this document we indicate the Reviewer's comments in *italic red*, while text that was changed or added in the manuscript is in blue. The line numbers mentioned here is in the manuscript.

*This study by Tiwari et al presents a novel utilization of a spatial metric SPAEF for calibrating and investigating the appropriateness of raw SNODAS snow water equivalent information in distributed hydrological modeling. The manuscript compares different calibration approaches including sequential, and multi-objective experiments using SWE and discharge, and a traditional single variable (discharge) based calibration, using different global optimization routines. The study is based in a snow-dominated catchment in Canada. The results indicate a better simulation of the spatial distribution of the SWE when SPAEF is used in calibration, with a more robust hydrological prediction.*

*I read the paper with interest given the similarity with my recent works. However, I have to say that this manuscript, in its current form, must undergo major revision to enhance the clarity for readers and for publication. The justification for this claim is based on the following review comments from my side.*

In response to the reviewer's comments on enhancing clarity for readers and for publications, we considered all the general as well as specific comments from all the reviewers and updated the manuscript accordingly.

*General comments:*

*1. The introduction section is not coherent and should be improved. I agree with Reviewer 1 that a major streamlining of the introductory section must be done. For instance, the description of SNODAS data can be brought together and shortened. Similarly, with the problems with hydrological models while using single discharge variable for calibration, leading to parameter compensations can be merged. Please avoid repeating statements. A suggestion would be to streamline the introduction section with explanations about the objective functions and optimization routine briefly explained in different sections. TSMM justification can be provided in the introduction section or conclusion, to avoid confusion for the results.*

We have undertaken the following revisions to the manuscript:

- Refined the abstract to summarize the key findings and contributions briefly.

- Streamlined the introduction by eliminating unnecessary details and relocating introductory content for better alignment to ensure that the key insights conveyed by the study is effectively communicated to the readers.

- Articulated the unique aspects of the study, particularly the innovative use of SPAEF and the impact of different calibration strategies.

- Moved the information about TSMM into the conclusion section.

*2. Consider putting a short novelty of the research in the abstract as well and shorten it with specific information from your research.*

Added a line emphasizing the novelty of the study in the abstract study "The novelty of this study is the implementation of SPAEF with respect to spatially distributed SWE for calibrating a distributed hydrological model. Lines 17-18

*3.The novelty of the study should be explicitly highlighted in the introductory section along with major findings. Multi-objective calibration is not a novel approach and has been identified to provide more reliable snow estimations with better or similar hydrological performance in different studies. However, the use of SPAEF for calibrating raw SNODAS data, as a novelty, should be highlighted comparing with past research employing similar approach with SNODAS or other data. The SPAEF metric used in other modeling approaches (Line 527) can be pulled to the introduction section.*

We agree with the reviewer here, we added the novelty of the study in the introduction section "The primary objective in this study is to introduce spatial calibration with SWE data using newly developed metric SPAEF for the calibration of the HYDROTEL hydrological model. We applied SPAEF in combination with other traditionally used objective functions. We conducted seven distinct calibration experiments, each employing a unique combination of objective functions. This allowed us to assess the trade-offs and robustness of these various calibration scenarios by evaluating their performance in terms of both streamflow and spatial SWE patterns. Notably, while SPAEF has been previously applied in studies involving evapotranspiration (Demirel et al., 2018) and soil moisture (Eini, Massari and Piniewski, 2023), this study uses SPAEF with SWE for the first time." Lines 109-115

*4.The usage of 'parameters' and 'variables' should be uniform as it can impart different meanings to the readers.*

Thank you for pointing this out. We have rectified the inconsistency in the usage of 'parameters' and 'variables' to ensure consistent understanding for the readers.

5.Please make a clear distinction between the spatial unit used for calibration. Is it RHHU or pixel based? If pixels, how is the spatial information (esp. SNODAS data) aggregated at RHHUs?

Clarification have been added in the methodology section.

While calibrating the model using SPAEF, spatial grid is utilized. The SWE values from SNODAS are in a 60 by 58 grid for Au Saumon watershed. For spatial calibration the mean SWE of each grid for the month of March are taken into account. This resulted in 60*58 spatially distributed SWE values for each calibration year. Subsequently, the SWE simulated by HYDROTEL for the same month (March) is transformed to match the same 60*58 spatial distribution of SWE values. These spatial patterns, representing SWE, are then calibrated using the SPAEF. Lines 264-268

6.How is the discharge extrapolated to the basin outlet?

The extrapolation of discharge to the basin outlet involves the summation of the cumulative flow throughout the basin area. The model calculates the flow from each cell toward the nearest stream or river segment, where it integrates with flows from adjacent cells (RHHUs). This integration results in an increasing discharge as the flow progresses downstream. The discharge at the basin outlet is the result of merging flows originating from different RHHUs within the basin. This includes both surface and subsurface runoff, integrating them to ascertain the overall outflow at the basin outlet. For a comprehensive understanding, kindly refer to (Fortin *et al.*, 1991). For this study the discharge is simulated with the kinematic wave equation on a daily basis.

7.The reasons behind calibrating the model solely against SWE distribution for March (peak accumulation) should be elaborated. Why not take the whole snow-season? How sensitive are the results when the calibration is done during the onset or the melt phases of the snow season? Since the study focuses on how raw SWE data can be used to constrain a hydrological model calibration, these sensitivity tests would further add value for other researchers and practitioners, for operational use.

Thank you for your insightful questions. We chose to calibrate the model solely against the SWE distribution for March because it is the month of maximum SWE for this watershed. This allowed us to maximize the use of available SWE information. However, we acknowledge the importance of considering the entire snow season and the potential impact of using different months on the calibration results. In response to your query about the sensitivity of the results during the onset or melt phases of the snow season, we conducted additional analyses using data from January and February. We added supplementary material with results when calibrated with other months (January and February). The results showed that SPAEF performs well with data from both these months. We agree with your suggestion that these sensitivity tests would add value for other researchers and practitioners, particularly for operational use. We believe that further research is necessary, with different watersheds and calibration period, to understand the performance of SPAEF during

the onset of snow accumulation and during the snowmelt period more accurately. A comment has been added in the discussion.

| | Calibrated with respect to SPAEF_March & NSE | Calibrated with respect to SPAEF_February & NSE | Calibrated with respect to SPAEF_January & NSE |
|---|---|---|---|
| NSE | 0.737 | 0.739 | 0.733 |
| KGE | 0.764 | 0.771 | 0.840 |
| RMSE Spatial | 39.38 | 51.90 | 50.23 |
| SPAEF wrt SNODAS Jan | 0.01 | 0.077 | 0.101 |
| SPAEF wrt SNODAS Feb | 0.157 | 0.201 | 0.181 |
| SPAEF wrt SNODAS March | 0.232 | 0.197 | 0.167 |

"For this study, March was selected for SPAEF calibration as it is the month with the highest SWE. Our objective was to leverage the maximum SWE information available during this period. However, we recognize that March, despite having the highest SWE, also overlaps with the snow melting period, which could potentially influence the calibration of our analysis. We performed additional analyses using data from January and February, and the results demonstrated that SPAEF performs well with data from both these months. We believe that further research is necessary, with different watersheds, to more accurately understand SPAEF's performance during the onset of snow accumulation and the snowmelt period. The detailed results of these additional calibrations can be found in the supplementary material, providing a comprehensive view of the model's performance."  Lines 516-523

*8.I agree with Reviewer 1 on the explanation of RMSE and NSE in the objective function section. This can be reduced. Also consider explaining why KGE specifically was used for validation.*

Removed sentences that were not necessary for the study. Since the model is calibrated using NSE, RMSE, and SPAEF, there was an interest in evaluating its performance using a different metric. Thus, the Kling-Gupta Efficiency (KGE) was chosen for validation to further assess the model's performance across an additional evaluation criterion.

*9.Given the coarser resolutions of the input drivers, how is the elevation dependent temperature and precipitation trends accounted for by Thiessen polygon method? This can have a detrimental effect on the simulation of snow accumulation and melt processes. Also, regarding the base model parameters used for comparison of model performance, did the cited research work with similar global model drivers? Consider giving a short explanation to formulate a sound basis for comparison.*

The Thiessen polygon method is selected to interpolate the meteorological data spatially, the values recorded at the station nearest to a specific cell is allocated to that respective cell. When Thiessen polygon is applied on a gridded data, the polygons generated outlines the area where a specific grid cell is the closest .Therefore, the Thiessen polygons align with the precipitation and temperature grid.

In this study, the base model serves as the starting point for the calibration experiments, rather than a comparison benchmark. All the experiments conducted in this study are compared to the standard experiment. In other words, all six experiments, each performed with a different objective functions, are compared to the standard experiment in which the model is calibrated with only the NSE.

"with vertical precipitation gradient of 1mm/100m and vertical temperature gradient of -1°C/100m." Lines 151-152

*10.Avoid reexplaining the calibration strategies, which makes the whole section longer.*

Thank you for the feedback. To address this comment, we implemented the recommended action by excluding redundant explanations of the calibration strategies in the article.

*11.Streamline the discussion section focusing on the results and corresponding relevant literature. Please avoid repetitions within the section and with the introduction. A short discussion on the uncertainties related to input data and SNODAS as compared to station observations would be better. Additional information on snow improving the hydrological understanding is not novel. Please focus on how your approach better represents the snow processes as well as parameter identifiability.*

The updated manuscript incorporates the suggestions provided. The discussion section has been revised, emphasizing the results with relevant literature, while avoiding repetition.

*12.Please specify the base conditions used for calibration (i.e choice of input data, model and the calibration variable) for strong conclusions like Concluding remarks #2, considering the study was done in only one catchment.*

Some remarks have been added at the end of the conclusion.

"The study, while conducted for a single watershed, contributes in our understanding of SPAEF's performance in hydrological modeling of snow-dominated watershed. However, it also reveals the need for further research. The utilization of different precipitation and temperature datasets as input data can significantly impact the performance of hydrological models. Variations in these datasets, which may arise from differences in data collection methods, spatial resolution, and temporal coverage, can affect the reliability and accuracy of hydrological predictions. The distinct characteristics of each watershed, including size, slope, altitude and land used can

have a substantial impact on the snow accumulation and melt processes. Therefore, it is essential to broaden this research to include different watersheds and various input data to validate and generalize our findings. Moreover, snow accumulation and melt do not occur uniformly throughout the year but happen in distinct periods. Our study focused on the month of maximum SWE (March), but the accumulation and melt periods of the snow season are both important. Future studies should consider different snow periods to gain a better understanding of SPAEF's performance." Lines 569-578

13. *How does this research differ from those who use bias corrected SNODAS information, particularly in terms of capturing the spatial distribution of snow cover and discharge simulation?*

Some remarks have been added in the discussion.

A number of researches have been done previously using bias corrected SNODAS and raw SNODAS information. King *et al.*,(2020) study revealed a significant enhancement in area melt estimates during the spring melt when utilizing bias-corrected SNODAS-SWE data compared to raw SNODAS estimates, which exhibited unrealistic melt volumes. The study's comparisons with in situ SWE measurements demonstrated that nonlinear bias-correction techniques notably improve the accuracy of SNODAS SWE estimates. Zahmatkesh (2019) showcased that bias-correcting SNODAS SWE significantly enhanced the accuracy of lumped models, contrasting with raw SNODAS SWE, which resulted in overestimated streamflow and peak flow values. A significant limitation in bias correcting SNODAS data lies in the absence of substantial data (Zahmatkesh *et al.*, 2019). Lines 451-458

Addressing this, the study utilized raw SNODAS data with SPAEF to evaluate hydrological model performance. Given that SPAEF has never been utilized with SNODAS, the focus was on assessing raw SNODAS data with a distributed hydrological model. The comparison of bias-corrected SNODAS data with a distributed hydrological model using SPAEF is open for future research endeavors. We encourage upcoming researchers to explore this area.

*Specific comments:*

*Line 2: Please make a clear and uniform distinction between snow 'parameters' and 'variables.*

The updated manuscript now presents a clear and consistent differentiation between snow 'parameters' and 'variables.

*Line 4: Full form of SWE at the first instance.*

The full form of SWE is already specified in line 2.

*Line 8: Remove 'approach'. 'calibration' should suffice.*

Removed.

*Line 9: Full form of SNODAS at the first instance.*

The full form of SNODAS (SNOw Data Assimilation System) is now provided.

*Line 14: Which "model performance"? Consider adding SWE and discharge simulation.*

Added.

*Line 5: 'hydrological events'*

Corrected.

*Line 56: Remove 'which'.*

As per the recommendations of both reviewers, the sentence related to TSMM has been omitted from the Introduction section and reformulated in the Conclusion section.

*Line 66: Add references here.*

Added.

*Lines 101 - 102: Consider rephrasing the sentence 'Ensuring….. streamflow forecasts'.*

The sentence is rephrased in updated manuscript.

*Line 136: 'spatialized'?*

Corrected.

*Line 150: Consider 'higher elevations' instead of 'hilly places".*

Corrected.

*Line 175: What does 'other modules' mean? Consider rephrasing.*

The word other is removed and the sentence is rephrased.

*Line 178: 'discharge is simulated' not 'model'*

Corrected.

*Line 202: Please check '2015-2020'.*

Corrected.

*Lines 214-215: Not necessarily. Add references or please rephrase.*

The sentence has been rephrased with the addition of references.

*Line 231: flow simulated at corresponding RHHU or at the outlet?*

Flow is simulated at the outlet. Since, there isn't a hydrometric station located at the outlet of the watershed, the discharge is calibrated where the discharge station is and the simulated flow at the corresponding RHHU.

*Lines 268 – 269: Please refer to general comments 5 on making a distinction between grid or RHHU. Better to make it clear beforehand in model description section.*

The clarification regarding the distinction between grid or RHHU has been incorporated into the updated manuscript as suggested.

*Line 280: 'run with some random parameter values' instead of 'ran with some random value'.*

Corrected.

*Line 306: It would be interesting to look at the SPAEF values for the base model as well.*

The SPAEF values for the base models have been included in the Table-4 .

*Line 318: Describe figure 4 in a better way than this. The current statement is confusing. Also please explain the significance of adding the 'rain-on-snow' description in this section.*

The statement has been revised in the updated manuscript to prevent confusion. The 'rain-on-snow' is added in the section as it allowed us to evaluate the model's performance during these events as rain-on-snow events during winter produce runoff, the model may tend to interpret these as streamflow.

*Table 4: Please add base model performance as well.*

Added in Table-4.

*Lines 327-328: Consider rephrasing this sentence about Figure 4.*

Rephrased.

*Lines 336-338: Consider rephrasing the sentence 'SPAEF….. year to year'. It will be interesting to see the SPAEF values for the onset and depletion phases of the snow season as well.*

Rephrased

Supplementary material has been included to observe the SPAEF values during both the onset and depletion phases of the snow season.

*Figures 3, 4, 5 and 7: Bigger font size please and also check the uniformity of the titles of these figures as well (esp. on top left). Consider plotting an average SWE line (simulated and observed) in these figures.*

The font sizes have been updated, ensuring uniformity in the figure titles within the updated manuscript. Figures 3, 4, 5, and 7 provide relevant information for the article. The decision to exclude the average SWE for both simulated and observed data was made to avoid potential confusion among readers.

*Lines 356 – 361: "relationship" or "comparison" ? Consider moving this section earlier or please put the experiment information in the caption in Figure 6.*

Corrected and the experiment information is included in the figure caption.

*Line 395: Please check if the calibration is done following 'NSE' based calibration.*

Thank you for pointing this out. The error has been rectified in the updated manuscript.

*Line 415: "under the considered model setting".*

Added.

*Line 432: Check SPAEFSWE*

Corrected.

*Lines 431-432: Explain more on this or remove this sentence about computational efficiency altogether.*

The sentence focusing on computational efficiency is removed in updated manuscript.

*Line 516: This sentence is confusing. Please rephrase.*

The sentence is rephrased.

*Line 518: 'simulation of complex processes' rather than 'complex processes.*

Corrected.

*Line 519: What is the high resolution defined here?*

*high temporal (ranging from daily to hourly) and spatial resolutions (up to a kilometer scale).* Lines 528

*Line 530: 'Snow variables' instead of 'snow parameters.*

Corrected.

*Overall, this research has some interesting findings on how raw SNODAS data can complement the existing hydrological modeling techniques, especially with a novel use of a SPAEF metric in a multi-objective calibration setting. However, the structure of the manuscript must be substantially revised for further acceptance. I would thus recommend further consideration of the manuscript after major revision.*

Demirel, M.C. *et al.* (2018) 'Combining satellite data and appropriate objective functions for improved spatial pattern performance of a distributed hydrologic model', *Hydrology and Earth System Sciences*, 22(2), pp. 1299–1315.

Eini, M.R., Massari, C. and Piniewski, M. (2023) 'Satellite-based soil moisture enhances the reliability of agro-hydrological modeling in large transboundary river basins', *Science of The Total Environment*, 873, p. 162396. Available at: https://doi.org/10.1016/j.scitotenv.2023.162396.

Fortin, J.-P. *et al.* (1991) 'HYDROTEL 2.1: user's guide.' Available at: https://espace.inrs.ca/id/eprint/921/1/R000315.pdf (Accessed: 4 November 2023).

King, F. *et al.* (2020) 'Application of machine learning techniques for regional bias correction of snow water equivalent estimates in Ontario, Canada', *Hydrology and Earth System Sciences*, 24(10), pp. 4887–4902. Available at: https://doi.org/10.5194/hess-24-4887-2020.

Zahmatkesh, Z. *et al.* (2019) 'Evaluation and bias correction of SNODAS snow water equivalent (SWE) for streamflow simulation in eastern Canadian basins', *Hydrological Sciences Journal*, 64(13), pp. 1541–1555.